# Sample-Efficient and Safe Deep Reinforcement Learning via Reset Deep Ensemble Agents

**Woojun Kim[1,*], Yongjae Shin[2,*], Jongeui Park[2], Youngchul Sung[2]**
[1]Carnegie Mellon University  [2]KAIST
woojunk@andrew.cmu.edu  {yongjae.shin, jongeui.park, ycsung}@kaist.ac.kr

## Abstract

Deep reinforcement learning (RL) has achieved remarkable success in solving complex tasks through its integration with deep neural networks (DNNs) as function approximators. However, the reliance on DNNs has introduced a new challenge called primacy bias, whereby these function approximators tend to prioritize early experiences, leading to overfitting. To mitigate this primacy bias, a reset method has been proposed, which performs periodic resets of a portion or the entirety of a deep RL agent while preserving the replay buffer. However, the use of the reset method can result in performance collapses after executing the reset, which can be detrimental from the perspective of safe RL and regret minimization. In this paper, we propose a new reset-based method that leverages deep ensemble learning to address the limitations of the vanilla reset method and enhance sample efficiency. The proposed method is evaluated through various experiments including those in the domain of safe RL. Numerical results show its effectiveness in high sample efficiency and safety considerations.

## 1 Introduction

With the remarkable success of deep learning in diverse fields such as image classification [10], the integration of reinforcement learning (RL) with deep learning, called deep RL, has been considered a promising approach to solving complex control tasks. In particular, using deep neural networks (DNNs) as function approximators enables the representation of state-action value function over a huge state-action space without requiring large storage as in tabular methods, which is often infeasible in complex tasks [12]. Despite its effectiveness, however, DNN-based function approximators in deep RL can lead to an overfitting problem called *primacy bias*, which was first investigated in Nikishin et al. [13]. Primacy bias means that DNN-based function approximators in RL may overfit to early experiences, impeding their ability as function approximators for subsequent later experiences. This phenomenon basically results from the use of an replay buffer in deep RL serving as a dynamic dataset to which experience samples are added during the training process so that experiences collected early in the training process are sampled more in training than those collected later [6]. Due to this primacy bias, it has been shown that performance tends to deteriorate as the replay ratio, defined as the number of updates per environment interaction, increases [6, 13].

To mitigate the primacy bias, Nikishin et al. [13] recently proposed a simple but effective method based on resetting. The reset method periodically resets either a portion or the entirety of a deep RL agent while preserving the replay buffer. This method enhances both the steady-state performance and sample efficiency by allowing the deep RL agent to increase the replay ratio, whereas the performance of the vanilla deep RL agent deteriorates with the increasing replay ratio. Despite its potential as a means to solve primacy bias, the reset method has a problem. That is, it causes periodic

---

*Equal contribution. Correspondence to: Youngchul Sung <ycsung@kaist.ac.kr>

collapses in performance immediately after the resets. The occurrence of periodic collapses is an undesirable side effect, which is an issue from the perspective of safe RL and regret minimization [13]. That is, performance collapse can lead to the violation of safety constraints, thereby limiting the applicability of the reset method to real-world RL scenarios such as autonomous vehicles. In addition, the reset method can be ineffective when the replay ratio is low, or the environment requires extensive exploration with balance with exploitation, for example, MiniGrid, as we will see in Sec. 4.

In this paper, we propose a new reset-based method that is safe—*avoiding performance collapses*—and achieves *enhanced sample efficiency*, by leveraging deep ensemble learning. In the proposed method, we first construct $N$-ensemble agents that adaptively integrate into a single agent based on the action value function, and then reset each agent in the ensemble sequentially. The adaptive integration of the ensemble agents prevents performance collapses after the reset by minimizing the probability of selecting the action from the recently-reset agent, also improves the sample efficiency by leveraging the diversity gain of the ensemble agents. We evaluate the proposed method combined with several deep RL algorithms such as deep Q-network (DQN) [12] and soft-actor-critic (SAC) [7], on various environments including Minigrid [5], Atari-100k [4], and DeepMind Control Suite (DMC) [17]. We show that the proposed method significantly improves the performance of the baseline algorithm in the considered environments. In addition, we apply the proposed method to a safe RL algorithm called Worst-Case SAC (WCSAC). The result demonstrates that the proposed method improves return performance while simultaneously ensuring the safety constraint. The main contributions of this work are summarized as follows:

• To the best of our knowledge, this is the first work that addresses the issue of primacy bias while simultaneously avoiding performance collapses, which can be detrimental in safety-critical real-world scenarios.

• We present a novel method that adaptively combines $N$-ensemble agents into a single agent, incorporating sequential resets for each ensemble agent, to effectively harness the diversity gain of the ensemble agents and prevent performance collapses.

• We empirically demonstrate that the proposed method yields superior performance compared with the base algorithm and the vanilla reset method on various environments. We also provide a comprehensive analysis of the underlying operations, including how the proposed method effectively prevents performance collapses.

• We provide a further experiment in the domain of safe RL, demonstrating that the proposed method surpasses the baseline approaches in terms of reward performance and effectively reduces the occurrence of safety constraint violations of the reset method during the training process.

## 2  Preliminaries and Related Works

We consider a Markov decision process (MDP). At each time step $t$, the agent selects an action $a_t$ based on the current state $s_t$ according to its policy $\pi(a_t|s_t)$. The environment makes a transition to a next state $s_{t+1}$ and yields a reward $r_t$ to the agent according to the transition probability $p(s_{t+1}|s_t, a_t)$ and the reward function $r(s_t, a_t)$, respectively. Through this iterative process, the policy $\pi$ is optimized to maximize the discounted return $R_t = \sum_{\tau=t}^{\infty} \gamma^\tau r_\tau$, where $\gamma \in [0, 1)$ is the discounted factor.

**Deep Ensemble Learning**  Deep ensemble learning has emerged as a promising approach, showing effectiveness in domains such as image classification and RL [3, 14]. It involves aggregating an ensemble of multiple DNNs to exploit their diversity and improve performance. RL with deep ensemble learning combines policies from multiple agents with the same architecture but with different initial parameters, leveraging their diversity to enhance robustness and efficiency. For example, Anschel et al. [3] proposed training and averaging multiple Q-networks for stabilization and improved performance. Lee et al. [11] employed an ensemble-based weighted Bellman backup technique that re-weights the target Q-value based on an uncertainty estimate from the Q-ensemble.

**Off-Policy RL**    Off-policy RL algorithms aim to optimize a target policy by using experiences generated by a behavior policy to increase sample efficiency compared with on-policy RL algorithms [13]. Off-policy learning typically uses a replay buffer to store experiences and use them to train the policy. One representative off-policy RL algorithm is DQN [12], which employs a DNN as a function approximator for the Q-function, defined as $Q_{DQN}^{\pi}(s_t, a_t) := \mathbb{E}_{\{a_t\} \sim \pi} \left[ \sum_{l=t}^{\infty} \gamma^{l-t} r_l | s_t, a_t \right]$. The

DNN parameters of the Q-function are trained to minimize the temporal difference (TD) error. Another example is SAC [7], which aims to maximize the cumulative sum of reward and entropy to improve exploration. SAC is built on the actor-critic architecture, consisting of a policy $\pi(\cdot|s_t)$ and soft-Q function, defined as $Q_{SAC}^\pi(s_t, a_t) := r_t + \mathbb{E}_{\tau_{t+1} \sim \pi} \left[ \sum_{l=t+1}^\infty \gamma^{l-t}(r_l + \alpha \mathcal{H}(\pi(\cdot|s_l)))|s_t, a_t \right]$. The policy is updated through soft policy iteration [7].

**Safe RL** Ensuring safety in RL is a crucial issue when applying RL to real-world applications such as autonomous vehicles and robotics. To address this issue, safe RL has emerged as an area of research that aims to learn policies satisfying safety constraints under a Constrained Markov decision Process (CMDP), which incorporates a cost function $c : \mathcal{S} \times \mathcal{A} \to \mathbb{R}$ to the conventional MDP framework. Recent safe RL research has focused on optimizing the return-based objective function while imposing constraints on the safety-related cost function [1, 9, 18]. For example, Yang et al. [18] introduced a SAC-based safety-constrained RL algorithm named WCSAC that aims to maintain the *Conditional Value-at-Risk (CVaR)* below a certain threshold. CVaR quantifies the level of risk aversion with respect to safety and CVaR for risk level $\alpha$ is defined as

$$\text{CVaR}_{\alpha_{risk}}(s, a) = Q_\pi^c(s, a) + \alpha_{risk}^{-1} \phi(\Phi^{-1}(\alpha_{risk})) \sqrt{\text{Var}_\pi^c(s, a)}, \tag{1}$$

where $\phi$ and $\Phi$ denote the PDF and the CDF of the standard normal distribution, respectively, and $Q_\pi^c(s, a)$ and $\text{Var}_\pi^c(s, a)$ respectively denote the conditional mean and the conditional standard deviation of the discounted sum of costs $C_\pi(s, a) = \sum_{t=0}^\infty \gamma^t c(s_t, a_t)$ given $s_0 = s$ and $a_0 = a$, which are called safety critic. WCSAC uses DNNs to parameterize the safety critic as well as the reward critic. Based on both the safety critic and the reward critic, the actor is trained to maximize the reward while not violating the risk level $\alpha_{risk}$.

**Primacy Bias and Resetting Deep RL Agent.** Enhancing sample efficiency in terms of environment interactions is a fundamental challenge in RL. One simple approach to improve sample efficiency is to increase the *replay ratio*, which is defined as the number of policy or value parameter updates per environment time step. In deep RL, however, the reliance on DNNs for function approximators can degrade performance as the replay ratio increases. This is because the deep RL agent is susceptible to overfitting on early experiences, which deteriorates the function approximation ability of DNNs for subsequent experiences. This phenomenon is referred to as primacy bias, first investigated in the context of RL by Nikishin et al. [13]. To overcome the primacy bias, Nikishin et al. [13] proposed a simple method based on increasing the replay ratio and periodically resetting either a portion or the entirety of the RL agent while preserving the replay buffer. It was shown that the deep RL algorithms employing both periodic resets and high replay ratios achieved impressive performance across several environments. In this line of research, D'Oro et al. [6] recently proposed fixing the number of updates for resetting, rather than the number of environment time steps, to maximally increase the replay ratio. This approach leads to more frequent parameter resets as the replay ratio increases, but results in improved sample efficiency in terms of interaction with the environment.

Although the existing reset methods mentioned above were shown to be effective in many tasks, certain limitations of the current reset methods exist. First, performance collapses are unavoidable immediately after executing the reset, which is not desired in the context of safe RL and regret minimization [13]. Second, the current reset method is ineffective for some off-policy RL algorithms such as DQN as well as on some environments requiring extensive exploration such as MiniGrid. Lastly, a high replay ratio is necessary to achieve the desired performance enhancement, which may not be supported in computing environments with limited resources.

## 3 Methodology

To overcome the aforementioned limitations and increase sample efficiency further, we propose a simple **R**eset-based algorithm by leveraging **D**eep **E**nsemble learning (RDE). The proposed method involves a sequential resetting process of ensemble agents that are adaptively integrated into a single agent. The algorithm is comprised of three main components: (1) the construction of ensemble agents, (2) the sequential resetting mechanism, and (3) the adaptive integration of the ensemble agents into a single agent. With RDE, we aim to achieve superior performance in various environments while simultaneously addressing safety concerns. The overall operation of RDE is shown in Fig. 1.

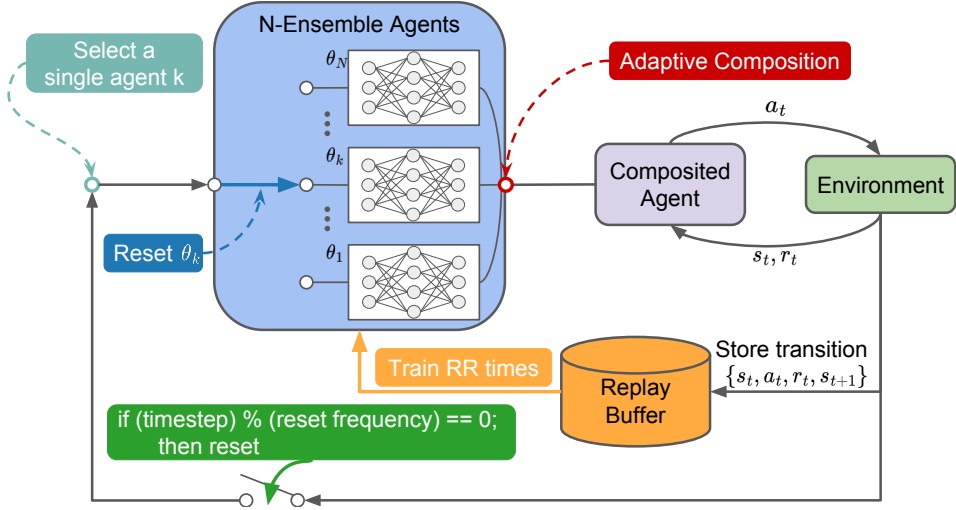

Figure 1: Overall diagram of RDE: We generate N ensemble agents with unique sets of initialized parameters. During the training phase, these ensemble agents are adaptively composited into a single agent that interacts with the environment. At every $T_{reset}$ time-step, a resetting mechanism operates by selecting a single agent $k$ and resetting all of its parameters $\theta_k$. Further details regarding the adaptive composition and sequential resetting mechanism can be found in Sections 3.2 and 3.1, respectively.

## 3.1 Sequential Resetting Ensemble Agents

We first construct $N$ ensemble agents, each with an identical neural network architecture but initialized with distinct parameters. In this paper, we denote the parameters of the $k$-th agent by $\theta_k$, where $k \in \{1, 2, \cdots, N\}$. For instance, in the case of SAC, $\theta_k$ includes all parameters for both the actor and critic of the $k$-th agent. Note that these ensemble agents are combined into a single agent that interacts with the environment. The integration of the ensemble into a single agent will be presented in Section 3.2.

Next, we perform the sequential and periodic resetting of the ensemble agents, while preserving the replay buffer. Specifically, every $T_{reset}$ time-step, we reset each parameter $\theta_1, \theta_2, \cdots, \theta_N$ in sequential order with time-step gap $T_{reset}/N$, returning back to $\theta_1$ after we reset $\theta_N$. Consequently, the existence of $N - 1$ non-reset agents at the time of each reset operation serves to mitigate performance collapses, whereas the vanilla reset method with a single agent inevitably encounters performance collapse after each reset. As the reset operation is applied to all ensemble agents, our method can still tackle the issue of primacy bias effectively.

## 3.2 Adaptive Ensemble Agents Composition

Despite the potential effectiveness of utilizing an ensemble of $N$ agents to mitigate performance collapses, the use of an ensemble alone is not sufficient. Suppose that we adopt a naive approach that selects one agent uniformly from the ensemble of the $N$ agents and follows the action that the agent generates. Then, performance collapses may still occur since the most recently initialized agent can be selected to generate untrained random actions or biased actions. Therefore, we need a method to judiciously integrate the ensemble into a single agent, considering that a lower probability should be assigned to the selection of the recently reset agent.

To achieve this, we propose the adaptive integration of the $N$-ensemble agents into a single agent that interacts with the environment, inspired by Zhang et al. [19]. The proposed integration method first generates a set of actions $\hat{a} = (a_1, a_2, \cdots, a_N)$ given a state $s$ from the policies $(\pi_{\theta_1}, \pi_{\theta_2}, \cdots, \pi_{\theta_N})$ of the ensemble agents. Subsequently, we select a single action from the set of $N$ actions with a probability distribution that is based on the action-value function. We assign a higher selection probability to the action with a higher action value, thereby reducing the chance that the most recently reset policy will be selected. Thus, the most recently reset policy is seldom selected immediately

after the reset operation, and the probability of selecting this element policy gradually increases as it is trained. The designed selection mechanism effectively mitigates performance collapses, as demonstrated in Sec. 4.3.

Let us explain the adaptive integration mechanism in detail. We design the probability distribution $p_{select}$ of selecting an action from the set of $N$ actions as a categorical distribution whose probability values are determined as

$$p_{select} = [p_1, p_2, \cdots, p_N] = \text{softmax} \left[ \hat{Q}(s, a_1)/\alpha, \hat{Q}(s, a_2)/\alpha, \cdots, \hat{Q}(s, a_N)/\alpha \right], \quad (2)$$

where $p_k$ denotes the probability of selecting the action from Agent $k$ and $\alpha$ is the temperature parameter. To deal with the scale of the Q-function, we set the temperature parameter as $\alpha = \max(\hat{Q}(s, a_1), \hat{Q}(s, a_2), \cdots, \hat{Q}(s, a_N))/\beta$ and adjust the coefficient $\beta$. Here, we choose $\hat{Q}$ to be the estimated action-value function of the agent that underwent a reset operation at the earliest point with respect to the current time step among the ensemble agents. For example, after the $k$-th agent is reset, the earliest reset agent is the $(k + 1)$-th agent (or the first agent if $k = N$). Then, $\hat{Q} = Q_{\theta_{k+1}}$ (or $Q_{\theta_1}$ if $k = N$). The underlying rationale for adopting this approach is that the estimated Q-value function of a recently reset network is inaccurate due to lack of training time. By leveraging the oldest Q-function, which provides a more precise estimation of the true action value function, we can effectively decrease the probability of selecting action $a_k$ that has a low return. We initialize $p_{select}$ before the first reset operation as $p_i = 1/N$ for all $i$.

In addition to its role in preventing performance collapse, our use of deep ensemble learning provides the potential to enhance performance through the diversity gained from ensemble agents. With ensemble agents initialized differently, we can effectively avoid getting trapped in local optima. Moreover, the adaptive integration allows the RL agent to balance the exploration-exploitation trade-off, in contrast to the vanilla reset method, which disrupts this balance by restarting from scratch and consequently losing its exploitation ability after a reset operation. The proposed method leverages the presence of $N-1$ non-reset agents, ensuring that the exploitation ability is preserved even after a reset. Thus, the proposed reset method is a more balanced approach in terms of exploitation-exploration trade-off, and still preserves some level of exploitation while exploring the environment.

In summary, the advantages of RDE are threefold: (1) it effectively prevents performance collapses, (2) it balances the exploration-exploitation trade-off through the adaptive ensemble integration, and (3) it improves sample efficiency by addressing the primacy bias and leveraging the benefits of deep ensemble learning. The pseudo-code of the overall algorithm is provided in Appendix A.

## 4   Experiments

### 4.1   Experimental Setup

**Environments**   We consider both continuous and discrete tasks including DeepMind Control Suite (DMC) [17], Minigrid[5], and Atari-100k[4] environments. Here, we consider 9 tasks in DMC, 26 Atari games, and 5 tasks in Minigrid. The details are provided in Appendix B.

**Baselines**   We built the vanilla reset method [13] and the proposed method, both of which were implemented on top of the base algorithms. We used SAC [8] and DQN [12] as the base algorithms in DMC and Minigrid/Atari-100K environments, respectively. For each environment, we compared three algorithms: the base algorithm (denoted as X), the vanilla reset method (denoted as SR+X), and the proposed method (denoted as RDE+X).

**DNN Architectures**   SAC employed an actor network and two critic networks, each consisting of three multi-layer perceptions (MLPs). For Atari-100k, DQN used a Convolutional Neural Network (CNN) for the first 3 layers and followed by 2 MLPs. For Minigrid, DQN employed 5 layers MLPs.

**Reset Depth**   Both the vanilla reset and our proposed method involve resetting parts or the entirety of DNNs. The degree of reset, referred to as the reset depth, is a hyperparameter that varies depending on the specific environment. We follow [13] in resetting the entire layers of DNNs in the SAC for the DMC environments. For the Minigrid environment, we observed that a higher reset depth is more advantageous for tasks requiring extensive exploration, which we will see in Sec. 4.3. The used reset depth in Minigrid and Atari-100k are provided in Appendix B.

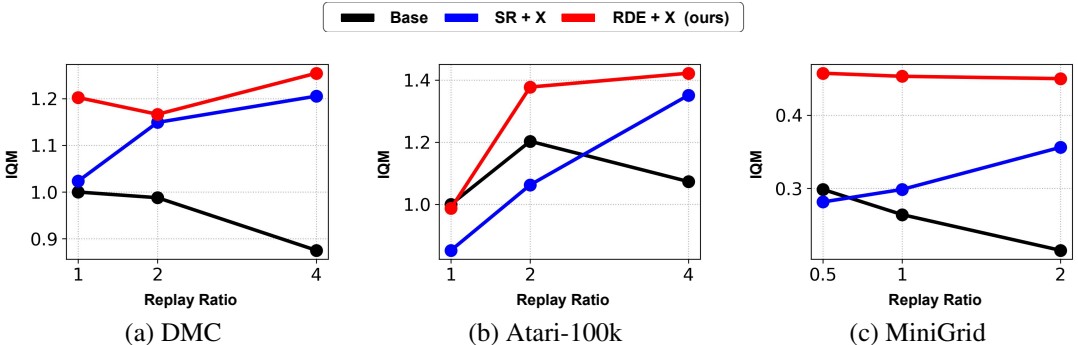

Figure 2: IQM results of the considered algorithms on (a) DMC, (b) Atari-100k, and (c) Minigrid. The y-axis in (a)-(c) represents IQM metric, based on the test return normalized by SAC with the replay ratio of 1, the test return normalized by DQN with the replay ratio of 1, and the test return, respectively. The x-axis denotes the replay ratio.

**Reset Frequency**    The reset frequency $T_{reset}$ is a hyperparameter that is fixed with respect to the number of updates, as proposed in D'Oro et al. [6]. Accordingly, for each environment, the reset frequency in terms of timesteps is a linearly decreasing function of the replay ratio. For example, let us assume that the reset frequency is $4 \times 10^5$ timesteps when the replay ratio is 1. Then, a replay ratio of 2 and 4 correspond to reset frequencies of $2 \times 10^5$ and $10^5$ timesteps, respectively. Note that the replay ratio can be smaller than 1. For instance, a replay ratio of 0.5 implies training DNNs every 2 timesteps. To ensure a fair evaluation, the reset frequency of each ensemble agent in RDE is equivalent to the reset frequency in the vanilla reset method. In summary, when the reset frequency with the replay ratio 1 in the vanilla reset is $T_{reset}^{v,rr=1}$, and the reset frequency with the replay ratio of $rr$ with $N$-ensemble agents is $T_{reset}^{v,rr=1}/(N \times rr)$. The reset frequency values for each environment is provided in Appendix B.

## 4.2    Performance Comparison

We first evaluated the performance of the proposed method in comparison with the base algorithm and the vanilla reset method by varying the replay ratio. The results on DMC, Atari-100k, and MiniGrid are shown in Fig. 2, using the interquartile mean (IQM) [2] performance metric. Note that the primacy bias is observed in the base algorithm, where its performance deteriorates as the replay ratio increases, as reported by Nikishin et al. [13] and D'Oro et al. [6]. While the vanilla reset method demonstrates better performance than the base algorithm, particularly with high replay ratios, our proposed method outperforms both baselines even with low replay ratios, as shown in Fig. 2.

The performance results of the proposed method and the baselines on the considered environments are shown in Fig. 3, where the considered tasks are `hopper-hop` and `humanoid-run` in DMC and `Fourroom` and `GotoDoor` in MiniGrid. It is seen that the proposed method not only yields superior performance to the baselines but also prevents performance collapses. In the `Fourroom` environment, the proposed method solves the game whereas both the base algorithm and the vanilla reset method fail to learn. The inherent loss of exploitation ability following a reset operation in the vanilla reset method prevents effective learning of this hard task. In other environments, as we expected, the vanilla reset method suffers from performance collapses. Especially in `humanoid-run`, there is a clear difference between our method and the vanilla reset method: immediately after a reset operation, our method sustains performance without a significant drop, whereas the vanilla reset method collapses to the initial performance.

More results on Atari-100k are provided in Appendix C.

## 4.3    Analysis

**Performance Collapse**    The results in Sec. 4.2 demonstrate the effectiveness of the proposed method in eliminating performance collapses. As aforementioned, the probability of selecting an action from the ensemble of $N$ actions is determined based on Eq. (2), which assigns a lower selection

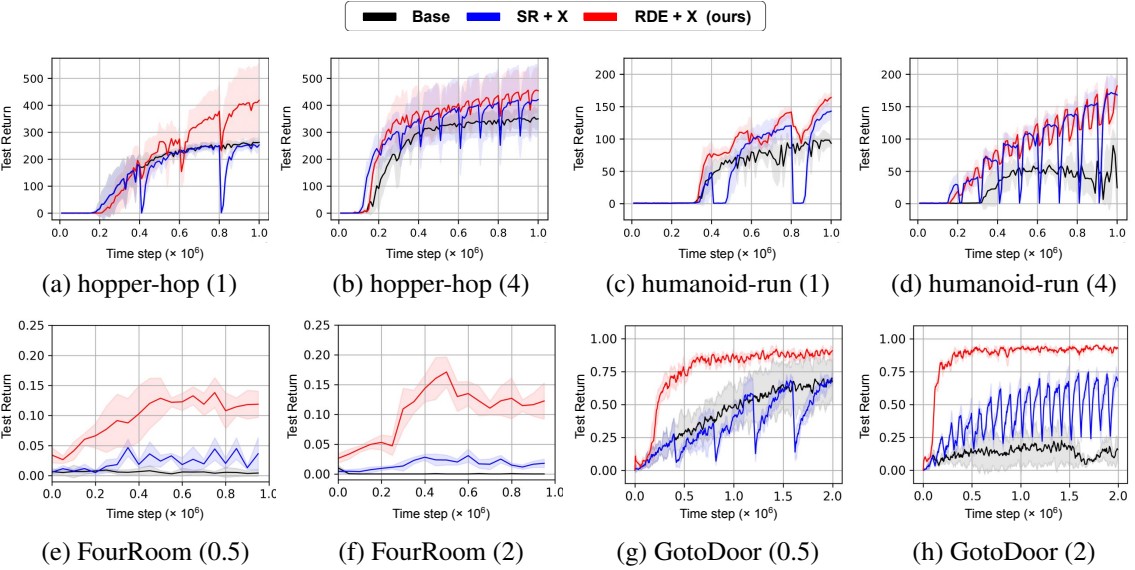

| (a) hopper-hop (1) | (b) hopper-hop (4) | (c) humanoid-run (1) | (d) humanoid-run (4) |

| (e) FourRoom (0.5) | (f) FourRoom (2) | (g) GotoDoor (0.5) | (h) GotoDoor (2) |

Figure 3: Performance comparison on `hopper-hop` and `humanoid-run` in DMC and `Fourroom` and `GotoDoor` in MiniGrid. Note that the number in parentheses indicates the replay ratio. The scale of the x-axis is $10^6$. Performances are averaged over 5 seeds.

probability to an action of a recently reset agent. This effectively prevents performance collapses, and the degree of this effect can be controlled by adjusting the parameter $\beta$. To investigate the impact of the parameter $\beta$, we conducted experiments on the `humanoid-run` task using RDE+SAC with

$N = 2$. We varied $\beta$ across $\{-10, 0, 50, 300\}$ and measured the corresponding empirical probability $p_1$, which denote the probability of selecting the action generated by $\theta_1$. In Fig. 4 (b), we observe that the empirical $p_1$ experiences a sudden drop immediately after resetting $\theta_1$. On the other hand, when $\theta_2$ is reset, $p_1$ jumps up, indicating the sudden drop of $p_2$. Note that the occurrence of performance collapses was nearly eliminated when $\beta$ was set to 300. This demonstrates the effectiveness of our adaptive integration in mitigating performance collapses

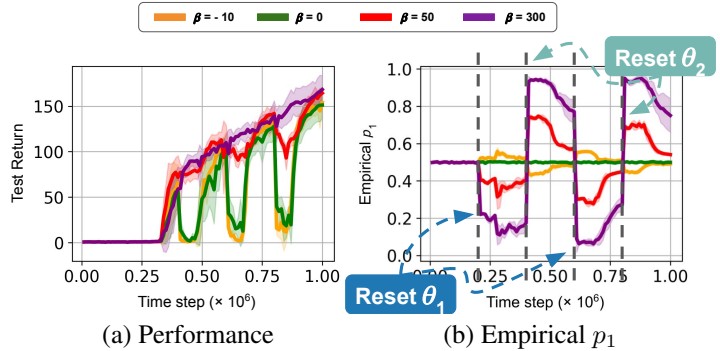

(a) Performance      (b) Empirical $p_1$

Figure 4: Performance and empirical probability of action selection from $\theta_1$ for RDE+SAC with varying $\beta$ values on `humanoid-run`.

by appropriately reducing the probability of selecting actions from recently-initialized agents. Furthermore, increasing the value of $\beta$ quickly adapt the values of $p_{select}$, leading to effective reduction of performance collapses.

**Reset Depth** The choice of reset depth is a crucial hyperparameter that significantly impacts the final performance of reset-based methods. We observed that a higher reset depth can provide an advantage in more challenging tasks that require hard exploration, whereas it may present a slight disadvantage in relatively easy tasks. We provide the performance of RDE+DQN with two different reset depths: *reset-part*, which resets the last two layers, and *reset-all*, which completely reset all layers. These evaluations were conducted on three MiniGrid environments, comprising two challenging environments, `FourRoom` and `SimpleCrossing9`, and one comparatively easier environment, `GotoDoor`. As shown in Fig. 5, *reset-part* exhibits inferior performance and instability compared with *reset-all* in two relatively challenging tasks. However, in two relatively easy tasks, the *reset-pt* performs marginally better than *reset-all*.

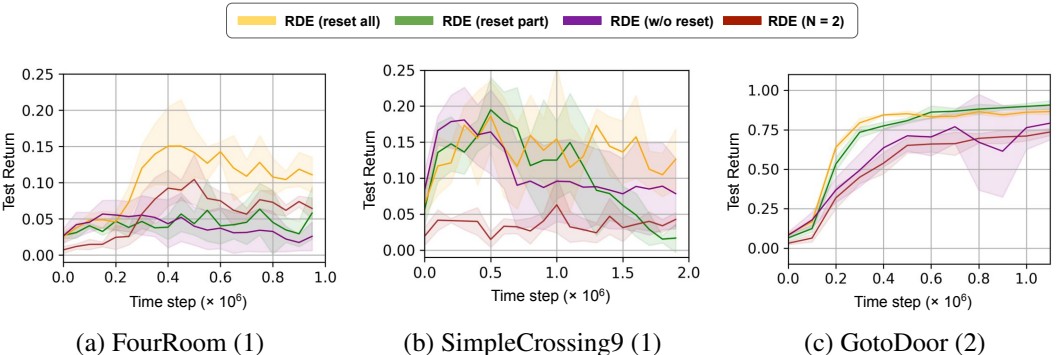

| (a) FourRoom (1) | (b) SimpleCrossing9 (1) | (c) GotoDoor (2) |

Figure 5: Ablation studies regarding reset depth, ensemble learning, and reset mechanism. Performances are averaged over 5 seeds.

**Ensemble and Reset Effect** The effectiveness of our proposed method is due to two key factors: ensemble learning and reset. To assess the contributions of these factors, we conducted two ablation studies by evaluating the performance of RDE by (1) varying the number of ensemble agents $N$, and (2) eliminating the reset mechanism. In the first ablation study, we compared the performance of RDE+DQN with $N = 4$ and $N = 2$ on the MiniGrid environments. The results, shown in Fig. 5, demonstrate that increasing the number of ensemble agents improves the performance of the proposed method. In the second ablation study, we examined the performance of RDE+DQN with and without reset. Here, the latter approach solely relies on deep ensemble learning. As shown in Fig. 5, the reset mechanism significantly contributes to the enhanced performance of the proposed method, especially in the FourRoom environment.

## 5 Experiment: Safe Reinforcement Learning

As previously mentioned, Safe RL aims to find a high-return policy while avoiding unsafe states and actions during training as much as possible. Ray et al. [16] formulated the problem as a constrained RL problem, where the agent incurs a cost when it visits a hazardous state. The goal is to maximize the expected return while ensuring that the cumulative cost remains below a predefined threshold. On the other hand, Jung et al. [9] considered maximizing the expected return while satisfying the probability of outage events. When applying the reset mechanism in such setting, a challenge arises because the deep RL agent loses all its knowledge about states that should be avoided after operating a reset. Consequently, until the safety critic is sufficiently retrained, the agent may continue to visit these undesirable states, thereby increasing the overall training cost. We expect that RDE with a slight modification taking into account both reward and cost can maintain the training cost low while benefiting from the performance gains associated with the reset operation still.

For safe RL, we adopted WCSAC, which maximizes the expected return while constraining the CVaR defined as Eq. (1), and modified the adaptive integration method proposed in Sec. 3.2. Specifically, we modified the probability of action selection in Eq. (2) by incorporating the cost function. The modified probability of adaptive action selection is now defined as $p_{select}^{safe} = \kappa p_{select} + (1-\kappa)p_{select}^c$, where $\kappa$ is the mixing coefficient and $p_{select}^c$ is given by

$$p_{select}^c = [p_1^c, p_2^c, \cdots, p_N^c] = \text{softmax}\left[-\hat{C}(s, a_1)/\alpha_c, -\hat{C}(s, a_2)/\alpha_c, \cdots, -\hat{C}(s, a_N)/\alpha_c\right], \quad (3)$$

where $\alpha_c = \max\{|\hat{C}(s, a_1)|, |\hat{C}(s, a_2)|, \ldots, |\hat{C}(s, a_N)|\}/\beta$. Here, we choose $\hat{C}$ to be the estimated CVaR value function that underwent a reset operation at the earliest point with respect to the current time step among the ensemble agent. Note that the sign of $\hat{C}(s, a)$ in Eq. (3) is inverted to prioritize actions with low CVaR values, as we aim to minimize the cost function.

### 5.1 Result

We compared our algorithm with WCSAC and SR-WCSAC, which incorporates the reset method into WCSAC, on the 3 environments in Safety-Gym. The details are provided in Appendix B.

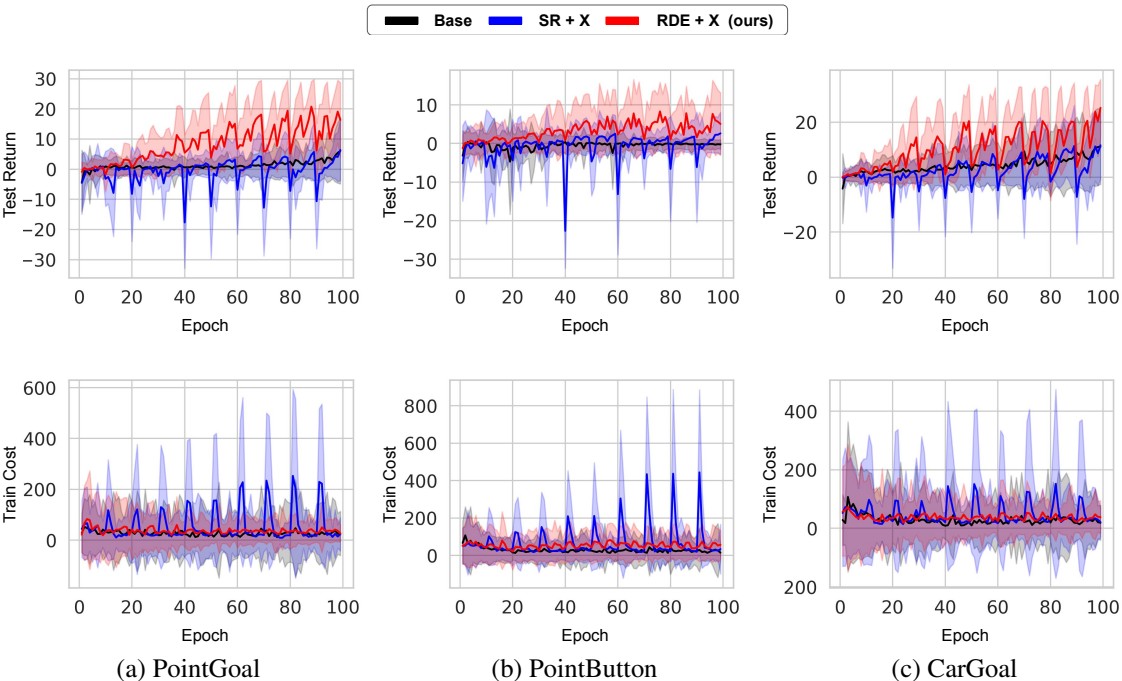

Figure 6: Performance Comparison in Safety Gym Environments. The first and second rows show the test returns and training costs, respectively. The performance metrics are averaged over five different seeds.

Fig. 6 shows the test return and the training cost. As expected, although the application of the simple reset method leads to a slight performance improvement compared with the baseline WCSAC agent, it also incurred a higher training cost. In particular, the soaring cost makes the naive reset method impractical in real-world scenarios concerning safety. On the other hand, our algorithm achieved superior test performance compared with the simple reset method, while significantly reducing the training cost. Note that the presence of non-reset agents and adaptive composition prevent cost from soaring after the reset operation.

## 6    Conclusion

We have proposed a novel RDE (**R**eset with **D**eep **E**nsemble) framework that ensures safety and enhances sample efficiency by leveraging deep ensemble learning and the reset mechanism. The proposed method effectively combines $N$ ensemble agents into a single agent to mitigate performance collapse that follows a reset. The experimental results demonstrate the effectiveness of the proposed method, which significantly improves sample efficiency and final performance, while avoiding performance collapse in various reinforcement learning tasks. In addition, in the context of safe reinforcement learning, our method outperforms the vanilla reset approach without incurring high costs, whereas the latter suffers from prohibitive training cost.

## Acknowledgments

This work was supported in part by Institute of Information & Communications Technology Planning & Evaluation (IITP) grant funded by the Korea government (MSIT) (No.2022-0-00469, Development of Core Technologies for Task-oriented Reinforcement Learning for Commercialization of Autonomous Drones) and in part by the National Research Foundation of Korea (NRF) grant funded by the Korea government (MSIT) (NRF-2021R1A2C2009143 Information Theory-Based Reinforcement Learning for Generalized Environments).

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

# A  Algorithm Pseudo-code

---
**Algorithm 1** RDE+$X$
---
    **Input:** Algorithm $X$, Ensemble size $N$, Reset frequency $T_{reset}$, Replay Ratio $RR$, Coefficient $\beta$,
    Initialize $N$-agents parameters, $(\theta_1, \theta_2, \cdots, \theta_N)$, and the replay buffer $\mathcal{D}$. Set index $k = 0$.
    **for** each episode **do**
        **for** each timestep $t$ **do**
            **for** $i = 1$ **to** $N$ **do**
                Collect action $a_t^i \sim \pi_i(a_t^i | s_t)$
            **end for**
            Calculate $p_{select}$ according to Eq. 2 and sample $a_t$ based on $p_{select}$.
            Execute $a_t$ and get $r_t$ and $s_{t+1}$. Store a transition to $\mathcal{D}$
            **for** $j = 1$ **to** $RR$ **do**
                Sample a random minibatch from $\mathcal{D}$
                Update $(\theta_1, \theta_2, \cdots, \theta_N)$ based on $X$
            **end for**
            **if** $t \% (T_{reset}/N) == 0$ **then**
                Reset $\theta_k$ and $k \leftarrow (k+1)\% N$
            **end if**
        **end for**
    **end for**

---

# B  Experimental Details and Hyperparameter

**1. DeepMind Control Suite.**    DeepMind Control Suite (DMC) is a collection of continuous control tasks that involve the manipulation of high-dimensional systems [17]. We considered nine distinct tasks of DMC, listed in Table 2. Our implementation is based on Stable Baseline3 [15] and the hyperparameters are provided in Table 1.

| Hyperparameters | Value |
|---|---|
| # of ensemble agents | 2 |
| Training steps | $1 \times 10^6$ |
| Discount factor | 0.99 |
| Initial collection steps | 5000 |
| Minibatch size | 1024 |
| Optimizer (all) | Adam |
| Optimizer (all) : learning rate | 0.0001 `humanoid-run` |
| | 0.0003 otherwise |
| Networks (all) : activation | ReLU |
| Networks (all) : n. hidden layers | 2 |
| Networks (all) : hidden units | 1024 |
| Initial Temperature | 1 |
| Replay Buffer Size | $1 \times 10^6$ |
| Updates per step (Replay Ratio) | (1, 2, 4) |
| Target network update period | 1 |
| $\tau$ | 0.005 |
| Reset Interval (gradient steps) | $4 \times 10^5$ |
| $\beta$ (action select coefficient) | 50 |

Table 1: Hyperparameters for RDE+SAC on DMC.

| Environment | Task |
|---|---|
| acrobot | swingup |
| cheetah | run |
| finger | turn_hard |
| fish | swim |
| hopper | hop |
| humanoid | run |
| quadruped | run |
| swimmer | swimmer15 |
| walker | run |

Table 2: The considered tasks of DMC

**2. Atari 100k.** Atari 100k is a benchmark that tests an agent's abilities by allowing it to interact with 100k environment steps (equivalent to 400k frames with a frameskip of 4) in 26 Atari games [4]. Each game has different mechanics, providing a diverse evaluation of the agent's capabilities. The benchmark imposes a restriction of 100k actions per environment, which roughly corresponds to 2 hours of human gameplay. The hyperparameters are provided in Table 3 and Table 4. Our implementation is based on Stable Baseline3 [15].

| Reset Interval | Environments |
|---|---|
| $4 \times 10^4$ | Assault, Asterix, Battle Zone, Boxing, Crazy CLimber, Freeway, Frostbite, Krull, Ms Pacman, Qbert, Seaquest, Up N Down |
| $8 \times 10^4$ | Alien, Amidar, Bank Heist, Breakout, Chopper Command, Demon Attack, Gopher, Hero, Jamesbond, Kangaroo, Kung Fu Master, Pong, Private Eye, Road Runner |

Table 3: Reset Interval in terms of the gradient step for each environment of Atari-100k

| Hyperparameters | Value |
|---|---|
| # of ensemble agents | 2 |
| Gray-scaling | True |
| Observation down-sampling | $84 \times 84$ |
| Frames stacked | 4 |
| Action repetitions | 4 |
| Reward clipping | [-1, 1] |
| Terminal on loss of life | True |
| Max gradient norm | 10 |
| Replay periode every | 1 step |
| Training steps | $1 \times 10^5$ |
| Discount factor | 0.99 |
| Initial collection steps | $1 \times 10^4$ |
| Minibatch size | 32 |
| Optimizer | Adam |
| Optimizer : learning rate | 0.0001 |
| Q network : channels | 32, 64, 64 |
| Q network : filter size | $8 \times 8, 4 \times 4, 3 \times 3$ |
| Q network : stride | 4, 2, 1 |
| Q network : activation | ReLU |
| Q network : hidden units | 512 |
| Replay Buffer Size | $1 \times 10^5$ |
| Updates per step (Replay Ratio) | (1, 2, 4) |
| Target network update period | 1 |
| Exploration | $\epsilon$-greedy |
| $\epsilon$-decay | $1 \times 10^4$ |
| $\tau$ | 0.005 |
| $\beta$ (action select coefficient) | 50 |
| Reset depth | last 2 layers: Amidar, Asterix Bank Heist, Freeway, Frostbite Gopher, Hero, Kangaroo, Pong last 1 layer: otherwise |

Table 4: Hyperparameters for RDE+DQN on Atari 100k.

**3. MiniGrid.**    Minigrid is a collection of goal-oriented tasks in 2D grid-world. The agent receives a sparse reward $R_1 = 10$ if the goal is achieved. The spare reward is penalized based on the number of interaction steps. In this paper, we considered 5 tasks of Minigrid including `FourRooms`, `SimpleCrossingS9N1`, `LavaCrossingS9N1`, `SimpleCrossingS9N1`, and `GoToDoor-8x8`. We now provide the brief introduction of each environment.

**FourRooms**   This environment is designed with four rooms and comprises of a single agent and a green goal. At the start of each episode, both the agent and the green goal are randomly placed within the four rooms. The goal of the agent is to navigate through the environment and ultimately reach the green goal.

**SimpleCrossingS9N1, LavaCrossingS9N1**   This environment is designed with two rooms that are blocked by obstacles, such as lava (for LavaCrossing) and walls (for SimpleCrossing). The objective is to successfully reach a goal while avoiding these obstacles. In LavaCrossing, the episode comes to an end if the agent collides with the obstacle, whereas in SimpleCrossing, the episode continues despite the collision.

**GoToDoor-8x8**   This environment is designed with a single room, four doors, and a single mission text string. The string provides instructions on which door the agent should reach.

**LavaGapS7**   This environment is designed with a single room, a strip of lava, and a green goal. The objective is to successfully reach the goal while avoiding the lava.

We provide the hyperparameters used in MiniGrid as follows.

| Hyperparameters | Value |
|---|---|
| $\epsilon$ | $0.9 \rightarrow 0.05$ |
| $\epsilon-$decay time step | $10^5$ |
| target update period | $10^3$ |
| Replay buffer size | $5 \times 10^5$ |
| Mini-batch size | 256 |
| Optimizer | RMSProp |
| Learning rate | 0.0001 |
| The maxmimum number of steps | 100 |
| Reset Interval (gradient steps) | $2 \times 10^5$ (GoToDoor, LavaCrossing, LavaGap) $1 \times 10^5$ (FourRooms, SimpleCrossing) |
| $\beta$ (action select coefficient) | 50 |

Table 5: Hyperparameters in MiniGrid.

**4. Safety-Gym.** Safety Gym presents a collection of environments having safety constraints. Within each of these environments, a robotic entity operates within a intricate setting with the objective of fulfilling a task while maintaining observance to predefined constraints governing its interactions with nearby objects and spatial regions. These environments incorporate separate reward and cost functions, which respectively describe task-specific objectives and safety prerequisites. We considered 3 environments including `PointGoal, PointButton`, and `CarGoal`. We now provide the brief introduction of each environment.

**PointGoal** A singular point robot is tasked with traversing toward a specific goal while effectively evading hazards. The point robot's mobility is confined to a 2-dimensional plane, facilitated by separate actuators enabling rotational motion and moving forward/backward.

**PointButton** This environment requires the point robot to sequentially press a series of goal buttons. These buttons, immobile in nature, are dispersed throughout the environment, compelling the agent to navigate and press towards the currently highlighted button, serving as the immediate goal.

**CarGoal** This environment introduces a robot of increased complexity, equipped with two independently driven parallel wheels and an additional freely rotating rear wheel. Effective coordination of both turning and forward/backward locomotion necessitates concurrent manipulation of both actuators. The objective of this environment coincides with that of the PointGoal environment.

| Hyperparameters | Value |
|---|---|
| # of ensemble agents | 2 |
| Epochs | 100 |
| Steps per epcoh | 20000 |
| Discount factor | 0.99 |
| Learning start steps | 500 |
| Minibatch size | 256 |
| Optimizer (all) | Adam |
| Optimizer (all) : learning rate | 0.001 |
| Networks (all) : activation | ReLU |
| Networks (all) : n. hidden layers | 2 |
| Networks (all) : hidden units | 256 |
| Initial Temperature | 1 |
| Entropy constraint | -1 |
| Replay Buffer Size | $1 \times 10^6$ |
| Target network update period | 1 |
| $\tau$ | 0.005 |
| Safety constraint | 25 |
| Risk level | 0.5 |
| Reset Interval (gradient steps) | $4 \times 10^5$ |
| $\beta$ (action select coefficient) | 50 |

Table 6: Hyperparameters in Safety Gym.

# C Experimental Results

We provide the entire results of DQN, SR+DQN, and RDE+DQN on Atari 100k and Minigird in Table 7 and Table 8, respectively. We report the per-environment learning curves of Minigird in Fig. 9. The learning curves of DMC are provided in Fig. 7 and Fig. 8.

| RR | 1 | | | 2 | | | 4 | | |
|---|---|---|---|---|---|---|---|---|---|
| Game | DQN | SR+DQN | RDE+DQN | DQN | SR+DQN | RDE+DQN | DQN | SR+DQN | RDE+DQN |
| Alien | 423.2 | 512.4 | 414.4 | 596.6 | 506.4 | 502.4 | 414.0 | **639.6** | 610.0 |
| Amidar | 46.8 | 43.2 | 47.8 | 54.6 | 58.2 | **68.2** | 31.6 | 66.4 | 55.2 |
| Assault | 438.8 | 354.5 | 409.1 | 409.9 | 369.2 | 431.8 | 372.1 | 455.3 | **462.0** |
| Asterix | 418.0 | 426.0 | 418.0 | 394.0 | 474.0 | 562.0 | 306.0 | 620.0 | **678.0** |
| Bank Heist | 14.0 | 13.6 | 16.8 | 23.2 | 27.2 | 21.2 | 14.4 | 26.8 | **33.6** |
| Battle Zone | 4040.0 | 3360.0 | 4040.0 | 2120.0 | 4520.0 | 7880.0 | 3840.0 | 7000.0 | **8240.0** |
| Boxing | 1.4 | -7.7 | -2.6 | 0.8 | -0.6 | 4.2 | **5.1** | 3.6 | 1.9 |
| Breakout | 16.1 | 6.7 | 16.1 | 19.6 | 15.0 | 21.2 | **23.8** | 20.8 | 19.5 |
| Chopper Command | 828.0 | 836.0 | 760.0 | **1324.0** | 1120.0 | 1000.0 | 1080.0 | 1024.0 | 1044.0 |
| Crazy Climber | 12472.0 | 16240.0 | 22556.0 | 22100.0 | 22216.0 | 25784.0 | 16028.0 | 25072.0 | **56324.0** |
| Demon Attack | 490.8 | 166.8 | 324.6 | **1492.4** | 184.4 | 652.4 | 1088.4 | 355.6 | 284.8 |
| Freeway | 15.1 | 7.2 | 4.0 | 10.9 | 6.4 | 16.8 | 14.8 | 7.9 | **21.2** |
| Frostbite | 233.6 | 158.4 | 197.6 | 206.4 | 206.8 | **348.0** | 116.4 | 264.4 | 271.6 |
| Gopher | 225.6 | 390.4 | 470.4 | 460.8 | 720.6 | 911.2 | 434.0 | **1169.6** | 1045.6 |
| Hero | 621.6 | 738.4 | 1698.6 | 1068.8 | 2725.6 | 2819.4 | 754.0 | 3073.2 | **3564.2** |
| Jamesbond | 68.0 | 70.0 | 126.0 | **178.0** | 50.0 | 138.0 | 140.0 | 92.0 | 78.0 |
| Kangaroo | 168.0 | 104.0 | 72.0 | 160.0 | 160.0 | 176.0 | 48.0 | **232.0** | 208.0 |
| Krull | 1905.2 | 2262.4 | **5325.6** | 2637.5 | 2460.4 | 1854.0 | 2533.6 | 3144.0 | 3374.0 |
| Kung Fu Master | 8264.0 | 5908.0 | 7256.0 | 6244.0 | 8216.0 | 7524.0 | 6008.0 | 7996.0 | **8284.0** |
| Ms Pacman | 790.0 | 769.6 | 609.6 | 907.2 | 832.0 | 831.6 | 868.4 | 954.8 | **1223.2** |
| Pong | -20.7 | -20.6 | -20.0 | -19.1 | -18.8 | -17.1 | **-14.4** | -16.0 | -16.8 |
| Private Eye | 20.0 | 2.1 | 44.0 | 44.0 | -69.1 | 64.0 | 0.0 | 40.0 | **83.5** |
| Qbert | 457.0 | 388.0 | 415.0 | 497.0 | 489.0 | 436.0 | 615.0 | 467.0 | **941.0** |
| Road Runner | 2288.0 | 576.0 | 1976.0 | 2288.0 | 2488.0 | 3180.0 | 1680.0 | 1924.0 | 3036.0 |
| Seaquest | 292.0 | 222.4 | 243.2 | 207.2 | 240.0 | 337.6 | 216.0 | 341.6 | **372.0** |
| Up N Down | 1396.8 | 1068.8 | 1734.4 | 1756.0 | 1769.2 | **2258.0** | 1662.4 | 1472.4 | 1503.6 |
| IQM | 1.000 | 0.852 | 0.987 | 1.203 | 1.063 | 1.377 | 1.073 | 1.351 | 1.422 |
| Mean | 1.000 | 0.603 | 1.092 | 1.323 | 1.050 | 1.695 | 1.181 | 1.641 | 1.902 |

Table 7: Results on Atari-100k

| RR | 0.5 | | | 1 | | | 2 | | |
|---|---|---|---|---|---|---|---|---|---|
| Game | DQN | SR+DQN | RDE+DQN | DQN | SR+DQN | RDE+DQN | DQN | SR+DQN | RDE+DQN |
| GoToDoor-8x8 | 0.709 | 0.71 | 0.911 | 0.544 | 0.659 | **0.944** | 0.159 | 0.684 | 0.929 |
| LavaCrossing | 0.035 | 0 | **0.248** | 0.016 | 0.02 | 0.215 | 0.029 | 0 | 0.237 |
| SimpleCrossingS9N1 | 0 | 0.012 | **0.186** | 0 | 0.013 | 0.159 | 0 | 0.014 | 0.137 |
| FourRooms | 0.002 | 0.03 | 0.148 | 0 | 0.072 | **0.159** | d 0 | 0.034 | 0.155 |
| LavaGapS7 | 0.747 | 0.655 | 0.793 | 0.761 | 0.729 | **0.793** | 0.674 | 0.706 | 0.791 |

Table 8: Results on MiniGrid

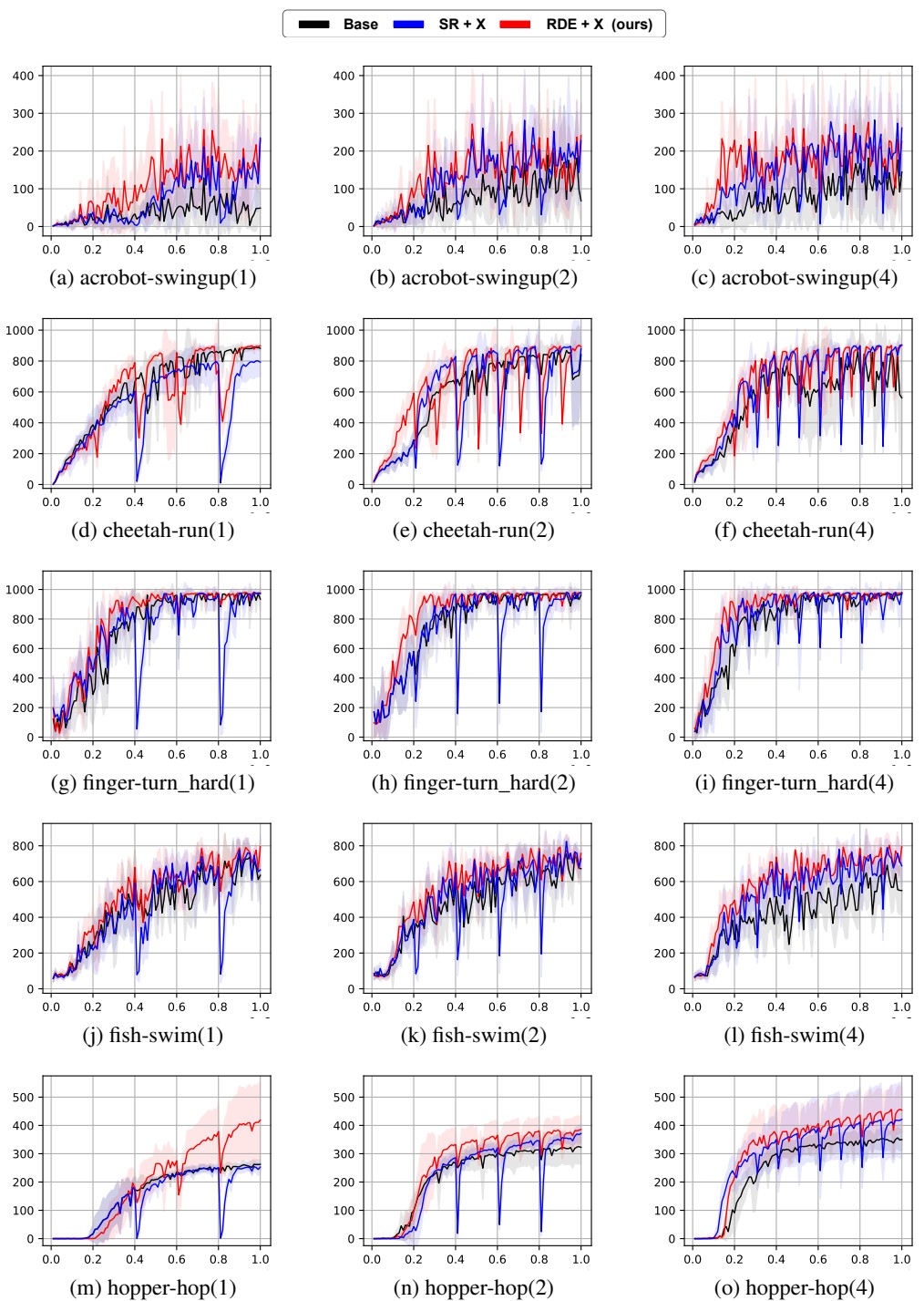

Figure 7: Per-environment performance in DMC with varying replay ratio values. Note that the number in parentheses indicates the replay ratio. The scale of the x-axis is $10^6$. Performances are averaged over 5 seeds.

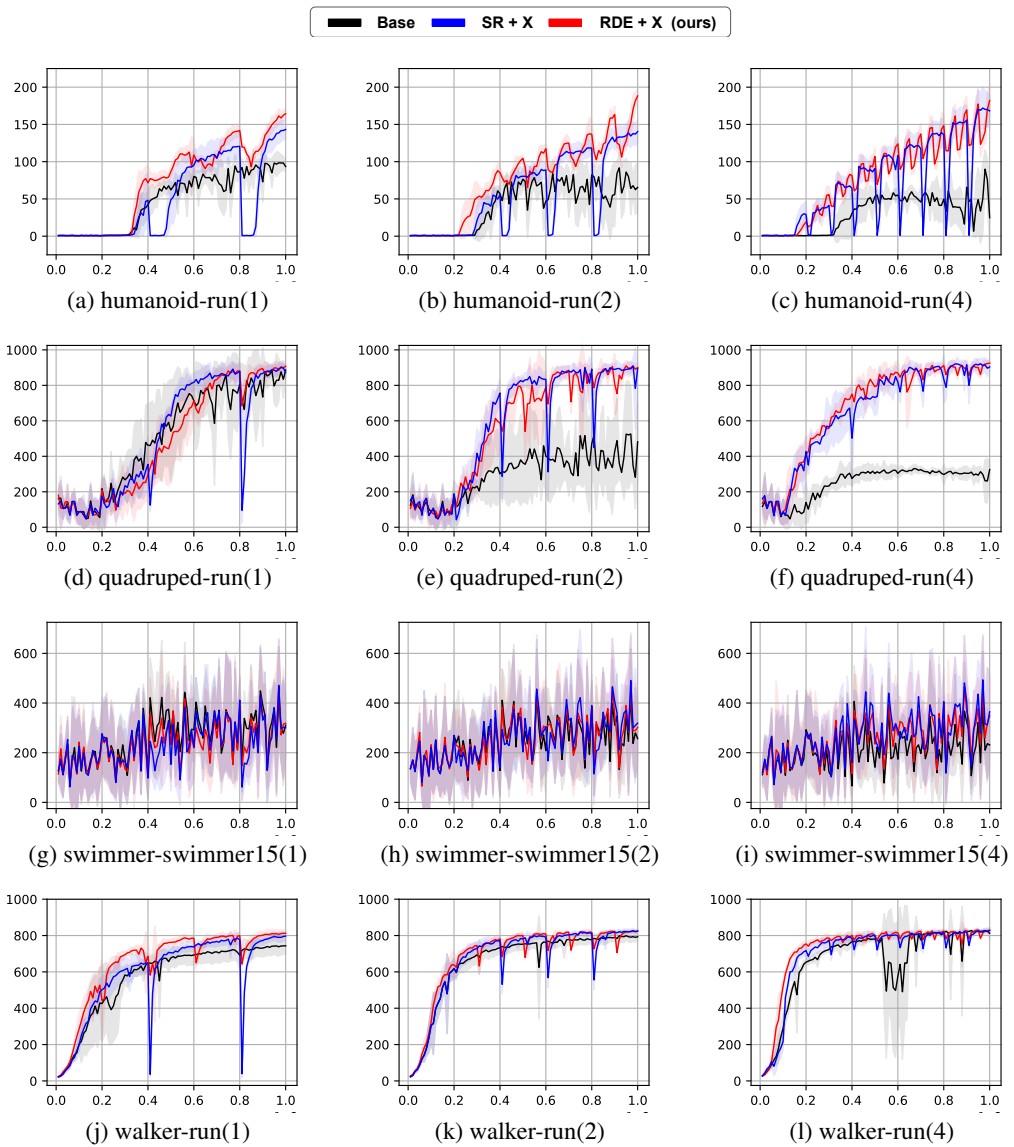

Figure 8: Per-environment performance in DMC with varying replay ratio values. Note that the number in parentheses indicates the replay ratio. The scale of the x-axis is $10^6$. Performances are averaged over 5 seeds.

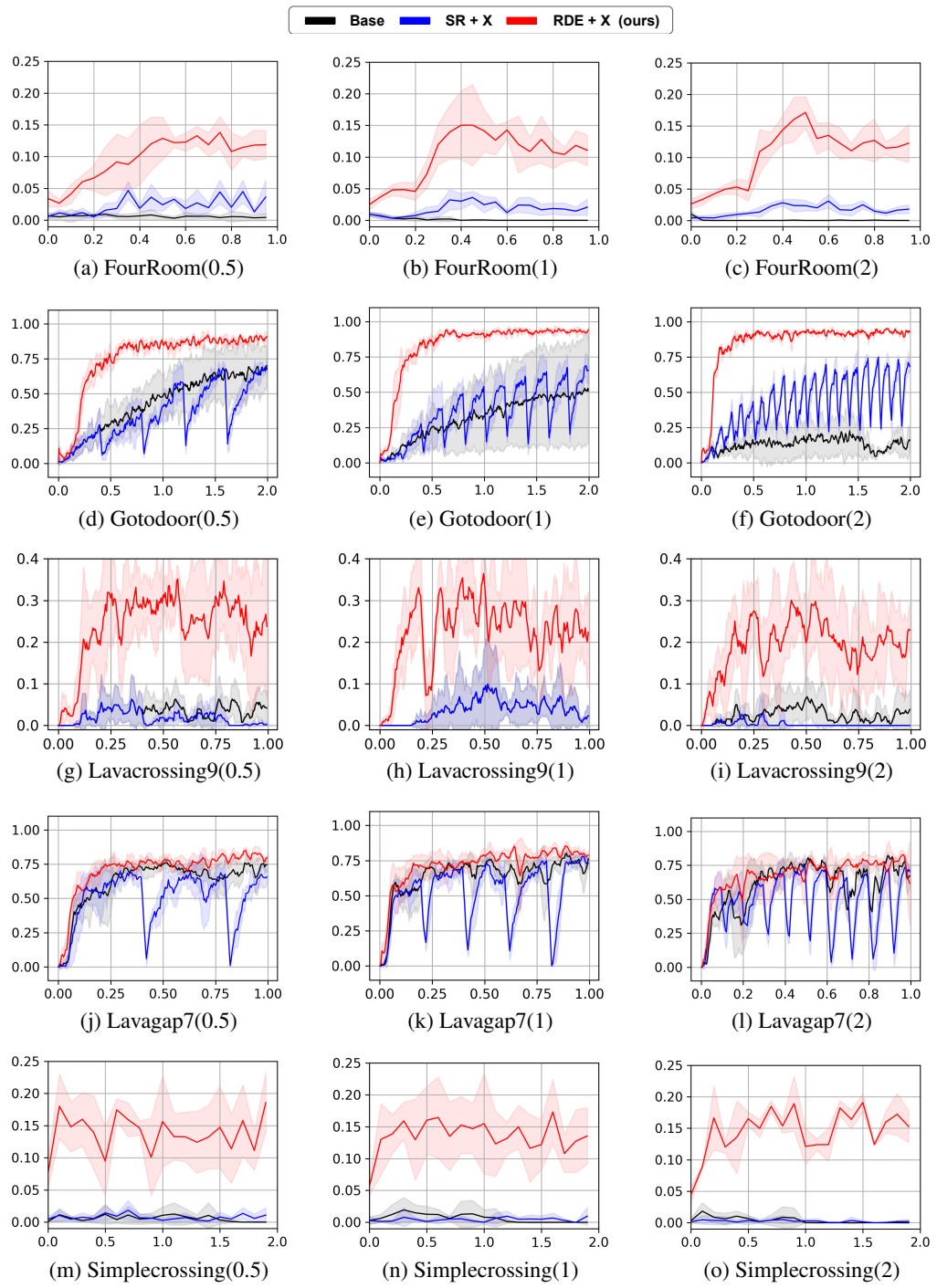

Figure 9: Per-environment performance in minigird with varying replay ratio values. Note that the number in parentheses indicates the replay ratio. The scale of the x-axis is $10^6$. Performances are averaged over 5 seeds.

# D  Reset depth for Continuous Environments

In Section 4.3, we investigated the impact of reset depth in Minigird environment. In order to investigate the effect of reset depth in continuous tasks, we conduct the additional experiments of RDE with two different depths: *reset-1*, which only resets the last layer, and *reset-all*, which entails a complete reset of all layers in the DMC environments. As shown in Fig. 10, it is observed that `reset-1` exhibits comparable performance to *reset-all* in the `cheetah-run`, `finger-turn_hard`, `hopper-hop`, `swimmer-swimmer15`, `walker-run` tasks. However, *reset-all* performs better than *reset-1* in the `acrobot-swingup`, `fish-swim`, `humanoid-run`, `quadruped-run` tasks. Furthermore, in the cases of `humanoid-run`, the performance of *reset-1* deteriorates with increasing replay ratio, suggesting that shallower levels of resetting render it more susceptible to primacy bias.

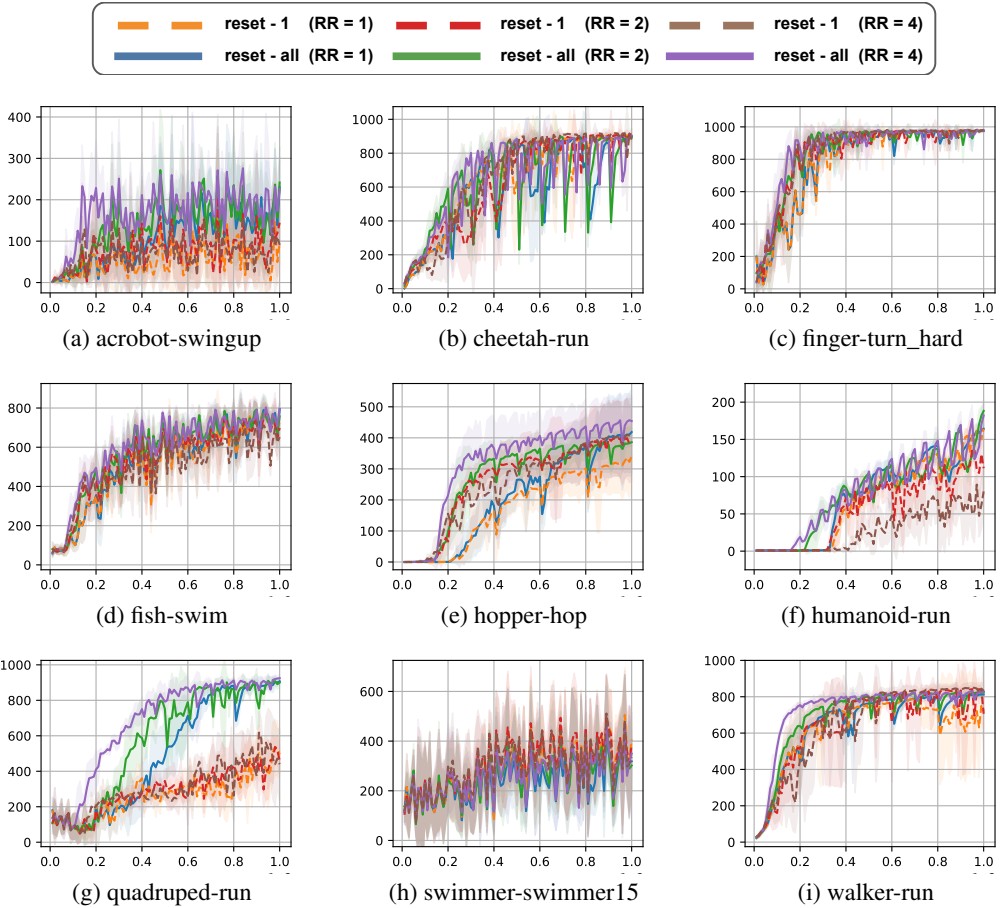

Figure 10: Per-environment performance in DMC with varying replay ratio values. Note that the number in parentheses indicates the replay ratio. The scale of the x-axis is $10^6$. Performances are averaged over 5 seeds.

# E  Further Experiments Regarding Reset Interval

The reset interval is an important hyperparameter in reset-based algorithms. We conduct performance comparisons by varying the reset interval, and the corresponding results are in Table 9. It is seen that RDE outperforms the vanilla reset method. To ensure fair comparison, we set the reset interval of one ensemble agent to match that of an SR agent. This is important because excessively frequent resetting can have a negative impact on the learning process, causing reset operations to occur before DNNs have fully recovered their performance. To illustrate this, we include additional experiments using the reset interval of $T_{reset}^{rr=1}/(N \times rr)$ for SR (vanilla reset) in DMC and Minigrid environments. The corresponding results are shown in Fig. 11 (indicated by the green line). Notably, the highly frequent vanilla reset (with the same number of reset operations as in RDE) performs worse, even more poorly than the base algorithm in the Minigrid environment.

| RR | 1 | | | 2 | | | 4 | | |
|---|---|---|---|---|---|---|---|---|---|
| $T_{reset}$ | $1 \times 10^5$ | $2 \times 10^5$ | $4 \times 10^5$ | $1 \times 10^5$ | $2 \times 10^5$ | $4 \times 10^5$ | $1 \times 10^5$ | $2 \times 10^5$ | $4 \times 10^5$ |
| SR+SAC | 1.00 | 1.08 | 1.02 | 1.00 | 1.15 | 1.15 | 1.02 | 1.13 | 1.21 |
| RDE+SAC | 0.99 | 1.10 | 1.20 | 1.02 | 1.10 | 1.17 | 1.02 | 1.16 | 1.25 |

Table 9: Results on DMC with varying reset interval

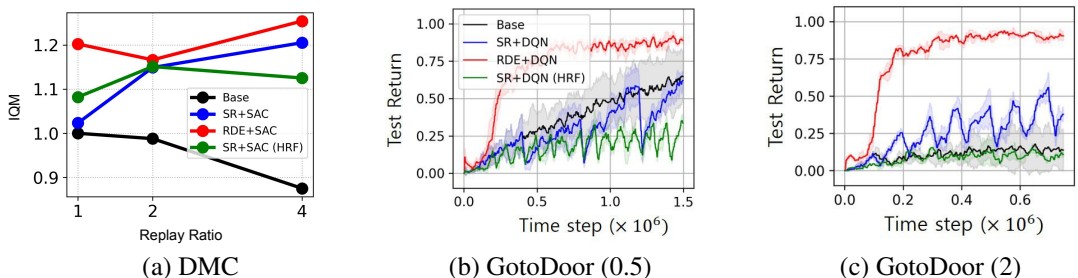

(a) DMC              (b) GotoDoor (0.5)              (c) GotoDoor (2)

Figure 11: The green line represents SR+X with an equal number of reset operations as RDE+X. (a) IQM metric normalized by SAC with a replay ratio of 1 on DMC. (b) & (c): Test performances on GotoDoor. High reset frequency (HRF) refers to the number of reset operations being the same as in RDE.

# F   Further Experiments Regrading # of Ensemble Agents ($N$)

We include additional experiments to confirm the effectiveness of ensemble learning. Increasing the number of ensemble agents, denoted as $N$, can lead to greater diversity gain. Additionally, the presence of $N - 1$ non-reset agents can aid in effectively mitigating performance collapses after resetting. Illustrated in Fig. 12, RDE with $N = 4$ demonstrates improved sample efficiency in the Minigrid environment. It is also observed in the Minigrid environment that the proposed method more effectively prevents performance collapse in comparison to RDE with $N = 2$.

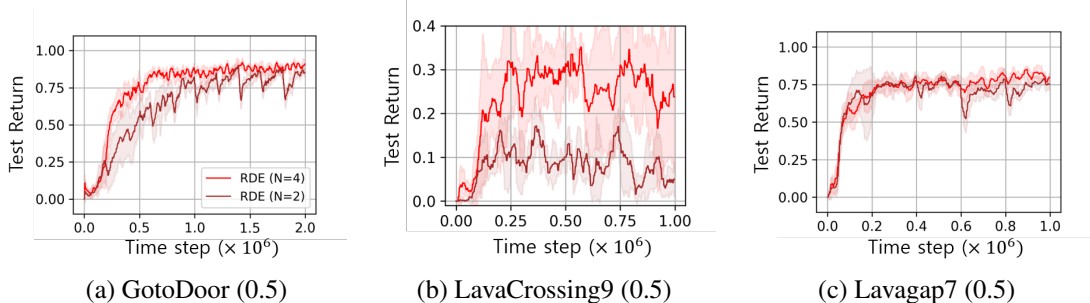

(a) GotoDoor (0.5)          (b) LavaCrossing9 (0.5)          (c) Lavagap7 (0.5)

Figure 12: Ablation study in MiniGrid environments with varying the number of ensemble agents, $N$. The red line and brown line represent RDE with $N = 4$ and RDE with $N = 2$, respectively.

# G   Limitations

Computational costs increase linearly as we raise the number of ensemble agents or the replay ratio. A limitation of our approach is the escalated computational cost due to ensemble agents. Nonetheless, it is important to highlight that, typically, the challenge in reinforcement learning lies more in sample efficiency, owing to the substantial costs tied to environment interaction, rather than computational intricacies. We are convinced that our method substantially enhances sample efficiency and safety, especially in environments with ample computational resources.

