# OpenReview forum: "Sample-Efficient and Safe Deep Reinforcement Learning via Reset Deep Ensemble Agents"
_NeurIPS.cc/2023/Conference — NeurIPS 2023 poster_

### Official Review · Reviewer_EaBR · 2023-06-26

**Soundness:** 3 good
**Presentation:** 3 good
**Contribution:** 2 fair
**Rating:** 5
**Confidence:** 4

**Summary:**

Overfitting in deep Q-learning agents is a recent topic of interest in the RL community, and several methods have been proposed to mitigate this problem, including data augmentation (DrQ), random ensembles (RedQ, DroQ), and resets (Nikishin et al.). This paper builds upon prior work on periodically resetting weights and addresses one of its core limitations: by resetting weights, an agent will perform poorly immediately after resetting, although it eventually recovers and often exceeds its performance before resetting. This paper proposes to learn an *ensemble* of agents and periodically reset only one of the agents at a time (in sequence). Experiments are conducted on tasks from DMControl, Minigrid, and Atari100k, and indicate that resetting with ensembles is effective at mitigating deterioration of performance immediately after a reset.

**Strengths:**

This paper is well written, considers an interesting problem, and experiments appear sound. The description of the proposed method is easy to follow, and especially Figure 4 is useful for understanding how the ensemble resetting works in practice.

**Weaknesses:**

- It would be useful to include more discussion on related works that seek to understand and mitigate overfitting in deep Q-learning besides resetting. There is a lot of literature in this area and I imagine that the authors are familiar with the literature, so I will refrain from mentioning any specific references (besides the methods mentioned in my summary) to remain impartial.

- I would like to see more ablations to get a deeper understanding of the trade-offs in ensemble resetting vs. prior works. How does sample-efficiency and performance drop change when the number of ensemble agents $N$ varies? How often should one reset agents? Is the rate of resets and number of agents dependent on the replay ratio? What is the computational cost of using additional agents vs. increasing the replay ratio? Would the proposed method benefit from using the ensemble in other ways as well, e.g. by using the RedQ trick for computing TD-targets? Addressing some or all of these questions would likely increase the impact of the work.

**Questions:**

I would like the authors to address my comments listed in "weaknesses". If the authors are not able to conduct some or all of the experiments required to answer my questions during the rebuttal, I'd like to see a discussion of what the authors would expect the results to be based on their experience.

**Limitations:**

There is sufficient discussion of limitations.

---

> ### Author Rebuttal · Authors · 2023-08-09
>
> **Regarding "Related works"**
>
> - We will include related works regarding overfitting problem in RL in the final manuscript, as the reviewer recommended.
>
>
> **Regarding "Sample efficiency and performance collapse with respect to the number of ensemble agents"**
>
> - We have conducted additional experiments concerning the number of ensemble agents, as detailed in the common response. As depicted in Figure 3 of the rebuttal PDF file, the results demonstrate that increasing the number of ensemble agents, denoted as $N$, improves both final performance and sample efficiency. This improvement is attributed to each agent being trained with experiences from a diverse set of ensemble agents, contributing to increased diversity. Moreover, the result shows that a higher $N$ leads to more efficient mitigation of performance collapse.
>
>
> **Regarding "Reset interval"**
>
> - We have included additional experiments concerning the reset interval, as outlined in the common response. It is seen that highly frequent resets can negatively impact learning. Consequently, determining an appropriate reset interval, a critical hyperparameter, is imperative. Importantly, our proposed method consistently outperforms the vanilla reset method across all considered reset intervals.
>
>
> **Regarding "Computational costs of increasing the number of ensemble agents and the replay ratio"**
>
>
> - Computational costs increase linearly as we raise the number of ensemble agents or the replay ratio. A limitation of our approach is the escalated computational cost due to ensemble agents. Nonetheless, it is important to highlight that, typically, the challenge in reinforcement learning lies more in sample efficiency, owing to the substantial costs tied to environment interaction, rather than computational intricacies. We are convinced that our method substantially enhances sample efficiency and safety, especially in environments with ample computational resources. We will include this discussion in the final manuscript.
>
>
> **Regarding "relationship between (the rate of resets, the number of agents) and the replay ratio"**
>
> - As discussed in our work, the reset method allows us to increase the replay ratio, resulting in improved performance. Therefore, a higher replay ratio allows for a higher rate of reset (more frequent resets). The number of ensemble agents is independent of the replay ratio, but we need to choose the appropriate number of ensemble agents and replay ratio within our computational budget, as both increase computational costs.
>
>
>
>
> **Regarding "Benefit from other ensemble learning methods"**
>
>
> - Ensemble learning offers a range of advantages, including improved exploration [4-1] and reduced estimation variance [4-2], depending on its purpose and approach. In our work, we leverage ensemble learning to prevent performance collapse in reset methods and attain diversity gains. Additionally, we concur that ensemble learning could yield further benefits, like diminished variance during Q-function estimation, as pointed out by the reviewer. While we recognize the potential impact of the recently reinitialized agent on ensemble learning benefits, this aspect could be explored in future work, prompting us to consider slight modifications.
>
> [4-1] Sunrise: A simple unified framework for ensemble learning in deep reinforcement learning
>
> [4-2] Averaged-dqn: Variance reduction and stabilization for deep reinforcement learning

---

> ### Author Response · Authors · 2023-08-19
> **An Invitation for Further Discussion**
>
> As the discussion stage is drawing to a close, we are eagerly awaiting your comments and suggestions.
>
> We believe that our responses, along with the additional experiments conducted during the rebuttal period, have effectively answered your questions, thereby enhancing the clarity of our work. We are grateful for the valuable suggestions and questions provided by the reviewer.
>
> Thank you for your valuable time and feedback.

---

### Official Review · Reviewer_fWBK · 2023-07-02

**Soundness:** 3 good
**Presentation:** 3 good
**Contribution:** 2 fair
**Rating:** 7
**Confidence:** 4

**Summary:**

This paper combines the resetting method proposed by (Nikishin et al., 2022) as a remedy to the primacy bias affecting deep RL algorithms with the use of ensembles of agents. The proposed RDE method, apart from generally improving performance, has the goal of minimizing the regret associated to a learning agent that uses periodic resets, mitigating the severeness of the performance drops it experiences right after a reset. To do this, without sacrificing too much on exploration and online data collection, the probability of executing an action in the environment is evaluated according to the oldest value function. Empirical results show benefits in using this approach, at both low and high replay ratio, in standard robotic locomotion and navigation tasks, as well as a safety domain.

**Strengths:**

**Originality**: despite the combination of resets and ensembles is not particularly original, the focus on developing a technique for leveraging their combination to avoid performance collapse while resetting is, to the best of my knowledge, new.

**Quality**: the quality of the work is generally good. The experiments cover a reasonable number of domains and the ablations mostly answer natural questions.

**Clarity**: the clarity of the writing is good. The paper would benefit from some small tuning to the presentation here and there, but the overall flow makes clear what the contribution is.

**Significance**: harnessing the performance benefits originating from the mitigation of the primacy bias while at the same time not incurring a cost in terms of regret is a worthy research direction which could be interesting to many practitioners and researchers.

**Weaknesses:**

**Major Concerns**
- If I understand it correctly from Algorithm 1, a different policy could be potentially selected at each step in the environment. This could conceptually create problems in terms of inconsistent behavior, since each policy will have to deal with the actions previously sampled from a potentially very different policy, and in terms of lack of "deep exploration", since this would harm temporally-consistent behaviors. A reader would benefit from a discussion or analysis of this aspect, to understand whether this is happening at all, or, if not, why that might be not happening in this kind of tasks of setting.
- The idea of combining periodic reset and ensembles of agents in continuous control has been explored in "Unleashing The Potential of Data Sharing in Ensemble Deep Reinforcement Learning" (Lin et al, 2022). I find the idea of the paper of using this combination to avoid performance drops to still be valuable, but discussing the relationship with that paper could better contextualize the contribution.

**Minor Concerns**
- Y and X labels are missing from all Figures in the paper, making it hard to parse the plots at first sight. For most plots, it is either quite easy to infer the quantities of interest, or they are explicitly mentioned in the caption, but it can still be very misleading for many readers.
- The paper would benefit from the extension of some of the ablations to more tasks. In particular, I find the performance of the ensemble-based approach without any resetting on top to be an important baseline to contextualize the results in Figure 3, and I think results concerning it would be a good addition to the Figure.

**Questions:**

I ask the authors to provide answers to my concerns expressed above.

**Limitations:**

The authors do not discuss any computational consideration resulting from their use of ensembles of agents. I encourage the authors to add such a discussion to the paper.

---

> ### Author Rebuttal · Authors · 2023-08-09
>
> **Regarding "Inconsistent behavior"**
>
>
> - It is entirely true that a different policy can be chosen at each time step. While this might result in inconsistent behavior, such inconsistency doesn't negatively affect exploitation since our method primarily relies on off-policy learning (the resetted policy is trained using experiences generated by the previous RL agent). Moreover, in terms of exploration, diverse policy behaviors can improve the process by generating new trajectories that have not been encountered before. In specific, we typically introduce Gaussian noise to actions in continuous action domains to encourage exploration, which can also potentially constrain the exploration of an RL agent. Through the utilization of policies from other ensemble agents, we expect that the adaptively composited agent can visit unexplored state-action spaces. We plan to include a discussion on this discussion in the final manuscript.
>
>
>
>
> **Regarding "Ensemble agents without reset"**
>
> - We have already compared our method to the ensemble-based approach without reset, and you can see the corresponding result in Figure 5 (purple line) of the main paper. This demonstrates that the combination of our ensemble learning and the reset method enhances performance.
>
>
>
> **Regarding "Relationship with a prior work"**
>
>
> As the reviewer mentioned, [3-1] also combines ensemble learning and the reset method. However, [3-1] primarily focuses on exploiting diversity gain by sharing data among ensemble agents, while our approach concentrates on both enhancing diversity gain and preventing performance collapses. The specific differences are as follows:
>
> - [3-1] employs parallel learning, training $N$ ensemble agents across $N$ corresponding environments. In contrast, RDE adapts and combines ensemble agents into a single agent within a single environment.
>
> - [3-1] resets ensemble agents simultaneously, whereas RDE performs a sequential reset to prevent performance collapse.
>
> - Additionally, we provide results in various RL domains, encompassing both discrete and continuous action spaces, as well as safe RL benchmarks.
>
>
> **Regarding "Figures"**
>
> - We will make more concrete revisions to the figures based on the comments from the reviewer.
>
>
>
>
> [3-1] Lin et al., "Unleashing The Potential of Data Sharing in Ensemble Deep Reinforcement Learning," arxiv, 2022

---

> > ### Comment · Reviewer_fWBK · 2023-08-16
> > **Thanks for the response**
> >
> > I am satisfied with the response from the authors. Provided that they will do the modifications to their paper (concerning figures, related work and other tweaks proposed in responses to other reviewers), my opinion is that the paper should be accepted.
> >
> > I am raising my score.

---

> > > ### Author Response · Authors · 2023-08-18
> > > **Thanks for the re-evaluation**
> > >
> > > We are thankful for the increased score.
> > >
> > > We will improve our paper based on the reviewers' comments.

---

### Official Review · Reviewer_UDiR · 2023-07-05

**Soundness:** 2 fair
**Presentation:** 2 fair
**Contribution:** 2 fair
**Rating:** 5
**Confidence:** 5

**Summary:**

The work proposes an extension to the resetting strategy proposed by Nikishin et al. The extension intends to mitigate the catastrophic performance collapse often observed for the simple resetting strategy while keeping the properties that help avoid the primacy bias.
To this end, the work makes use of ensembling techniques, such that the policy/value network of an RL agent is not completely reset, but can fall back to different checkpoints.
The method is evaluated on a variety of environments and a further extension is presented that enables application in safety critical systems.

**Strengths:**

* The presented method cleverly combines ensembling with resetting to improve RL agents.
* It seems general enough that it should be usable as a plug and play method for a broad variety of RL agents.
* The work mostly easy to follow
* It is demonstrated that the method is flexible and can incorporate auxiliary information, such as safety-critical information to provide a safer method than the vanilla resetting one

**Weaknesses:**

The presentation could be improved:
* A lot of the discussed "preliminaries" seem irrelevant for the content of the paper. For example, the paragraph on "Off-policy RL." seems not necessary and the content in "Primacy Bias and Resetting Deep RL Agent." is largely a repetition of the introductory text.
* Algorithm 1 is never discussed in the text and feels wholly redundant with Fig. 1. This half page might be used to show more experimental results.
* Algorithm 1 is not consistent with the text. From the Algorithm it looks like every ensemble member is reset every $N\times T_{reset}$ time-steps and not $T_{reset}/N$ as stated in line 149.
* Lines 178 - 180 are concerned with expressing that the "oldest Q-function" is used to normalize in the selection mechanism. This can be expressed in much clearer terms "oldest Q-function" as is shown in line 183.
* In line 195 it is claimed that RDE effectively prevents performance collapses, however I disagree with this wording. It can mitigate it to some extent, but the experiments clearly show that there is still performance collapses happening.
* The description of Figure 2 is wholly confusing. The long sentence explaining what the y-axis is showing is expressed in a very convoluted way.
* In figure 2c it should not be possible that the performance of the baseline "Base" is below 1 when using an RR of 1.
* Design decisions are often not well enough explained.
* Figure 1 does not explain what RR is
* The paragraph heading in line 204 should be "Baselines & RL Agents" not "Baselines & DNN Architectures"
* Line 246 claims that there is no significant drop on the humanoid-run example. The reward for the RDE method drops from 150 - 100. This represents a significant drop in my opinion since 1/3 of the performance is lost.
* The choice for designing the selection mechanism is very unclear. The discussion about using the oldest Q-function in the selection mechanism seems more to point to only using the oldest Q-function. Indeed, this seems to be supported in the experiments and should have been at least an ablation. Additionally the initial paragraph of Section 3.2 seems to also point to only using the oldest Q-function
* It is claimed that a $\beta$ set to 50 "nearly eliminated" performance collapses (line 270). However, the performance collapse in the Figure is again the 1/3 loss in performance.
* Section 5 feels like an added afterthought. I don't see why it warrants an own section. It repeats some of the discussion of safe RL from the preliminaries section. Instead the "safe" selection mechanism should have been discussed in Section 3 and then only the results should be a subsection of Section 4.
* Where does the value for the reset frequency $4\times 10^5$ in line 221 come from?

The experiments seem to have an unfair comparison and are likely not reproducible:
* All details about how hyperparameters were determined seems to be missing
* The value of $N$ is never stated. Only from the plots can it be assumed to be 2
* The ablation does not take into account all confounding factors and should be redone
* It is claimed that using a reset frequency for the individual members being equal to the single network case is fair. I fail to see how this is fair. This seems to just give benefit to the RDE method since it can benefit $N$ times as much from resetting.
* Without stating how hyperparameters were set of the methods, it seems like an arbitrary comparison of the methods.
* N is not really ablated and it's not clear how the ensemble increases computational overhead.

The description of an MDP is wrong. An MDP is an abstract representation of an environment. The MDP does not consist of an agent.
The bounds for the discount factor in line 74 should be $\gamma \in [0,1]$ not $\gamma \in [0,1)$. It is totally valid to have undiscounted cases.

Overall, the work would need significant rewriting and an overhaul for the experiments for me to accept it. I am very doubtful that this can be done in the time-frame of a rebuttal.

**Questions:**

How would PBT style methods compare to resetting strategies. Are they doing some form of resetting?

**Limitations:**

Limitations of the method have not been discussed. An obvious limitation is that ensembles likely will increase the computational overhead. For example, in line 131 it is said that the simple resetting strategy often requires high replay ration and therefor more resources. This should likely be worse for the presented method and the conducted experiments are not convincing enough to show that the novel method would require fewer resources.

---

> ### Author Rebuttal · Authors · 2023-08-09
>
> **Regarding "Presentation"**
>
>
> - The subsection on 'Off-policy RL' is necessary in our work for two reasons. Firstly, the reset methods depend on an off-policy algorithm, as a recently reinitialized RL agent needs training using experiences generated by the previous RL agent. Secondly, this section allows us to introduce the base algorithms used in our experiments (DQN and SAC) and essential definitions for our work, such as the value function, which plays a key role in the adaptive integration mechanism (Equation 2).
>
> - In the 'Safe RL' subsection of Preliminaries, we introduce the definition of safe RL and provide an example of a safe RL algorithm (WCSAC). In contrast, Section 5 explores a specific challenge that arises when applying the reset method to safe RL domains: the rapid increase in safety costs. We then describe how we incorporate the proposed method into WCSAC to tackle this challenge and present the corresponding results. Thus, we believe both sections are necessary to establish the background and motivation of our work.
>
> - We agree that there is overlap between the content of "Primacy Bias and Resetting Deep RL Agent" and the Introduction. We intend to streamline and simplify the duplicated portions within "Primacy Bias and Resetting Deep RL Agent" for the final manuscript.
>
> - We plan to relocate Algorithm 1 to the Appendix and offer a comprehensive explanation of it in the final manuscript. Furthermore, we will adjust the reset frequency in Algorithm 1 to $T_{reset}/N$.
>
> - We apologize for the confusion regarding Fig. 2. The y-axis of Fig. 2 represents the IQM metric of test return. In Fig. 2 (a) and (b), we normalized the test return using the base algorithm with a replay ratio of 1, while in Fig. 2 (c), we utilized the unnormalized test return (which is why the "base" baseline performance is below 1). To ensure consistency, we will make the necessary revision to display the normalized value in the final manuscript.
>
> - The value of $4\times 10^5$ is one example.
>
>
> **Regarding "Prevention of performance collapse"**
>
> - As discussed in Section 4.3, we can address the degree of performance collapses by adjusting $\beta$. In the rebuttal PDF file, we have incorporated a result from RDE using a higher $\beta$ on the Humanoid-run environment. The result, illustrated in Figure 4 of the rebuttal PDF file, shows that RDE with $\beta=300$ avoids performance collapse, affirming the effectiveness of our method. In addition, RDE with $\beta=50$ still performs better than the base algorithm even if performance collapses exist.
>
>
> **Regarding "Experiments and Hyperparameters"**
>
> - The hyperparameters have been tuned within the range of values used in prior work [2-1]. Furthermore, to ensure fair comparison, we established the hyperparameters to be consistent; for example, both SR and RDE employ the same number of reset operations per agent (see common response). Regarding the reset interval, we have included a performance comparison by adjusting it, as mentioned in the common  response for the reviewer's information. The range of hyperparameters will be detailed in the final manuscript.
>
>
> - We have mentioned the value of $N$ in the Appendix; however, we will relocate it to the main body. For Atari-100k and DMC, we used $N=2$, while for Minigrid, we used $N=4$. Notably, we conducted an ablation study on the number of ensemble agents, $N$, as outlined in "Ensemble and Reset Effect" in Section 4.3. During the rebuttal period, we also included the additional ablation study on $N$, and the corresponding results are shown in Figure 3 of the rebuttal PDF file.
>
>
> **Regarding "Reset frequency"**
>
> - We have discussed the concept of fair comparison related to reset frequency (interval) in the common response. To summarize, the reset operation of an individual agent in RDE should be the same with that of the SR agent, considering that the recovery times for DNNs of the same size tend to be similar. Notably, we noticed a decrease in performance when the reset frequency for an SR agent corresponds to the frequency at which ensemble agents are reset. Please see the common response and Figure 2 in the rebuttal PDF file (the green line represents SR with the same number of reset operations as ensemble agents).
>
>
>
> **Regarding "PBT style method"**
>
> - Population-Based Training (PBT) involves training networks in parallel with diverse parameters and hyper-parameters. It periodically evaluates performance, selects the best hyper-parameters, and distributes them to other learners for training. While both PBT and our method involve multiple networks, it's important to note that PBT is not directly related to reset methods. Reset methods, on the other hand, focus on reinitializing the parameters of a deep neural network to avoid overfitting to early experiences and facilitate convergence towards the global maximum (or minimum).
>
>
> **Regarding "Using the oldest Q-function in adaptive composition"**
>
> - The rationale behind utilizing the oldest Q-function is that the estimated Q-value function of a recently reset network can be unreliable due to limited time for recover its performance. We have conducted an ablation study regarding this, and the corresponding result is shown in Figure 5 of the rebuttal PDF file in the common response. The result indicates that utilizing the recently reinitialized (newest) Q-function for adaptive composition yields poorer performance due to inaccurate cumulative return estimation. The first, second, and third oldest Q-functions exhibit similar performance since they all recover their performance to approximate the cumulative return. From a conservative standpoint, we believe employing the oldest Q-function is the most suitable choice for our method.
>
>
>
> [2-1] E. Nikishin et al., "The primacy bias in deep reinforcement learning," ICML 2022

---

> > ### Comment · Reviewer_UDiR · 2023-08-11
> > **Rebuttal Reponse (Increased score)**
> >
> > Thank you very much for the detailed response and the many additional experiments. My concerns and issues have been mostly addressed and I am much more positive towards to presented work. I increase my score form 3 to 5.
> >
> > You stated in the rebuttal that you tuned the hyperparameters. How was this done? A grid search? Random search? How much tuning is required to get a good performance of the method? Did you use the same tuning budget for all methods?
> >
> > I am still concerned that the work requires substantial rewriting which might warrant a resubmission. Since other reviewers have not raised that point though I will discuss that in the reviewer discussion.

---

> > > ### Author Response · Authors · 2023-08-16
> > > **Response to Reviewer UDiR**
> > >
> > > We thank the reviewer for the positive consideration of our work.
> > >
> > > - We conducted grid-based hyperparameter tuning, focusing on a range of values employed in previous research. This stems from the fact that our method is an extension of prior work [2-1], designed to prevent performance collapse and harness ensemble gains. Particularly in the environments utilized in the prior research, such as DMC, we initiated our search for appropriate values based on the parameters used in that previous work. For instance, in [2-1], a value of $2 \times 10^4$ was employed for $T_{reset}$ in the DMC environment. Consequently, we fine-tuned the values to ${1 \times 10^5, 2 \times 10^5, 4 \times 10^5}$.
> > >
> > >
> > > - Additionally, in environment such the MiniGrid environment, which had not been explored in prior studies, we performed grid searches for hyperparameter tuning. For example, in the case of tuning $T_{reset}$ for the Minigrid environment, we considered values within the set ${2.5\times 10^4, 5\times 10^4, 1\times 10^5, 2 \times 10^5 }$. Once we determine the appropriate recovery time for DNNs, achieving strong performance becomes more manageable. In addition, as mentioned in the common response, we set the hyperparamter considering the fairness.
> > >
> > >
> > >
> > > - Note that we used the same budget for the common hyperparameters tuning.
> > >
> > >
> > > - As stated in the previous comment of the rebuttal, we will enhance our presentation in accordance with the reviewers' feedback. We are confident that the final manuscript will be well-structured and comprehensive.

---

### Official Review · Reviewer_6LEC · 2023-07-06

**Soundness:** 3 good
**Presentation:** 3 good
**Contribution:** 2 fair
**Rating:** 4
**Confidence:** 4

**Summary:**

This paper proposes a novel reset-based method that leverages deep ensemble learning to address the primacy bias issue in deep reinforcement learning. The authors construct N-ensemble agents and reset each ensemble agent sequentially to prevent performance collapses and improve sample efficiency. The proposed method is evaluated through experiments on various environments, including safe RL scenarios, and the results demonstrate its effectiveness and potential for real-world applications.

**Strengths:**

1. The paper addresses an important issue in deep reinforcement learning, namely primacy bias, and proposes a novel method to mitigate its effects. This is valuable as primacy bias can lead to overfitting and performance deterioration, which affects the applicability and efficiency of deep RL algorithms.
2. The use of deep ensemble learning in the proposed method is innovative and practical. Deep ensemble learning has shown effectiveness in domains such as image classification and RL, and leveraging the diversity gain of ensemble agents can enhance performance and prevent performance collapses.
3. The paper provides a comprehensive analysis of the proposed method, including its underlying operations and how it effectively prevents performance collapses. This analysis adds clarity and depth to the understanding of the proposed method.

**Weaknesses:**

1. The paper stated that RDE was conducted for tasks with safety requirements, but it only did one experiment on safe RL benchmark. Thus the paper lacks effective validations for RDE on violations. Simply stating that RDE does not cause performance collapse in general continuous control tasks cannot explain its role in safe RL.

**Questions:**

1. Can you show more experimental results on safe RL benchmark?

**Limitations:**

The paper stated that RDE was conducted for tasks with safety requirements, but it only did one experiment on safe RL benchmark. The paper has no negative social impacts.

---

> ### Author Rebuttal · Authors · 2023-08-09
>
> **Regarding "Results on safe RL benchmark"**
>
> - As described in the common response, we have conducted two additional experiments on the safe RL benchmark to show the effectiveness of the proposed method. The corresponding results are presented in Figure 1 of the rebuttal PDF file. These results clearly indicate that RDE outperforms WCSAC in both test performance and cost. As demonstrated earlier in the main paper, RDE addresses a critical issue of the naive reset method in the safe RL domain—the rapid increase in safety costs during training. Summarizing the aforementioned additional results along with the result on safe RL domain in the main paper, it becomes evident that the proposed RDE not only ensures safety but also enhances sample efficiency.

---

> ### Author Response · Authors · 2023-08-19
> **An Invitation for Further Discussion**
>
> As the discussion stage is drawing to a close, we are eagerly awaiting your comments and suggestions.
>
> We believe that our responses, along with the additional experiments conducted during the rebuttal period, have effectively addressed your concern—additional experimental results on the safe RL benchmark. If you have any remaining concerns, we would greatly appreciate the opportunity to engage in a productive conversation with you.
>
> Thank you for your valuable time and feedback.

---

### Author Rebuttal · Authors · 2023-08-09

We thank all reviewers for their valuable comments.

In this paper, we propose a novel method that incorporates ensemble learning into the resetting method to harness diversity gain and mitigate performance collapse. We provide various experiments on both standard and safety RL benchmarks, as well as ablation studies that demonstrate how the proposed method mitigates performance collapses. We believe the proposed method can contribute to practical RL algorithms by addressing sample efficiency and safety, which are critical challenges in RL.

In response to the reviewers' comments, we conducted additional experiments and included the corresponding results in the rebuttal PDF file. Referring to the PDF file, we present our common response to the major concerns raised by the reviewers below:


**1. Two Additional Experiments on Safe RL Benchmark**


In addition to the results on Safexp-PointGoal1-v0, we have included results on the SafexpPointButton1-v0 and SafexpCarGoal1-v0 environments. The results indicate that RDE consistently outperforms WCSAC in terms of both test performance and cost across all considered environments, as shown in Figure 1 of the rebuttal PDF file. As RDE improves final performance while minimizing safety cost regret, we believe that RDE can significantly contribute to the safe RL domain.


**2. Further Experiments Regarding Reset Interval**


- The reset interval is an important hyperparameter in reset-based algorithms. We have conducted performance comparisons by varying the reset interval, and the corresponding results are as follows:


| RR |\| | 1 |  | \| | 2 | | \| | 4 | |
| :------------ | :------------ | :-------------: | :-------------: | :------------ | :-------------: | :------------: | :------------ | :------------: | :-------------: |
| $ T_{reset} $ | \| | $ 2 \times 10^5 $ | $ 4 \times 10^5$ | \| | $ 2 \times 10^5 $ | $ 4 \times 10^5 $ | \| | $ 2 \times 10^5 $ | $ 4 \times 10^5 $ |
| **SR+SAC** | \| | 1.08 | 1.02 | \| | 1.15 | 1.15 | \| | 1.13 | 1.21 |
| **RDE+SAC** | \| | 1.10 | 1.20 | \| | 1.10 | 1.17 | \| | 1.16 | 1.25 |

It is seen that RDE outperforms the vanilla reset method.

- To ensure  fair comparison, we set the reset interval of one ensemble agent to match that of an SR agent. This is important because excessively frequent resetting can have a negative impact on the learning process, causing reset operations to occur before DNNs have fully recovered their performance. To illustrate this, we've included additional experiments using the reset interval of $T_{reset}^{rr=1}/(N\times rr)$ for SR (vanilla reset) in DMC and Minigrid environments. The corresponding results are shown in Figure 2 of the rebuttal PDF file (indicated by the green line). Notably, the highly frequent vanilla reset (with the same number of reset operations as in RDE) performs worse, even more poorly than the base algorithm in the Minigrid environment.



**3. Further Experiments Regarding # of Ensemble Agents (N)**

We have included additional experiments to confirm the effectiveness of ensemble learning. Increasing the number of ensemble agents, denoted as $N$, can lead to greater diversity gain. Additionally, the presence of $N-1$ non-reset agents can aid in effectively mitigating performance collapses after resetting. Illustrated in Figure 3 of the rebuttal PDF file, RDE with $N=4$ demonstrates improved sample efficiency in both Atari-100k and the Minigrid environment. It is also observed in the Minigrid environment that the proposed method more effectively prevents performance collapse in comparison to RDE with $N=2$.


**4. Further Experiments Regarding Performance Collapse**

We have presented a result from RDE with a higher $\beta$ on the humanoid-run environment to determine whether RDE can completely prevent performance collapse. The corresponding result is depicted in Figure 4 of the rebuttal PDF file. It is observed that RDE with $\beta=300$ does not experience any performance collapse.



**5. Further Experiments Regarding Adaptive Composition of Ensemble Agents**


We outlined the reasoning for employing the oldest Q-function in the main paper, considering that the recently reinitialized Q-function may not provide accurate estimations. To demonstrate this, we conducted an ablation study using alternative Q-functions, such as the 2nd oldest, 3rd oldest, and newest Q-functions. Figure 5 of the rebuttal PDF file illustrates that utilizing the newest Q-function leads to notably worse performance. From a cautious perspective, using the oldest Q-function for the adaptive composition is the most appropriate decision for our approach.


**6. Presentation**



- We will enhance the presentation of our paper using the insightful comments provided by the reviewers. For instance, in response to Reviewer UDiR's feedback, we plan to relocate Algorithm 1 to the Appendix and provide a comprehensive explanation of its details to address the concern that Algorithm 1's content may appear redundant with the information presented in Fig. 1. Furthermore, we intend to refine the figures to offer a clearer representation.


- The computational overhead almost linearly increases as $N$ increases, which is a limitation of our work, and we will include it in the final manuscript.

---

### Decision · Program_Chairs · 2023-09-21

**Decision:**

Accept (poster)

**Comment:**

This paper proposes a solution for primary bias in deep RL which does not have the adverse safety issues of the plain resetting solution. Most of the reviewers agree that this is an interesting work.

The only reviewer who rejected the paper asked for more safe RL experiments and the authors added 2 more experiments during the rebuttal. Another reviewer had concerns about the writing. However, given the very positive review by the reviewer fWBK, I recommend an acceptance and encourage the authors to improve their writing in the final version of the paper.